# Dynamic emotional states shape the episodic structure of memory

Mason McClay [1], Matthew E. Sachs [2,3] & David Clewett [1] ✉

Human emotions fluctuate over time. However, it is unclear how these shifting emotional states influence the organization of episodic memory. Here, we examine how emotion dynamics transform experiences into memorable events. Using custom musical pieces and a dynamic emotion-tracking tool to elicit and measure temporal fluctuations in felt valence and arousal, our results demonstrate that memory is organized around emotional states. While listening to music, fluctuations between different emotional valences bias temporal encoding process toward memory integration or separation. Whereas a large absolute or negative shift in valence helps segment memories into episodes, a positive emotional shift binds sequential representations together. Both discrete and dynamic shifts in music-evoked valence and arousal also enhance delayed item and temporal source memory for concurrent neutral items, signaling the beginning of new emotional events. These findings are in line with the idea that the rise and fall of emotions can sculpt unfolding experiences into memories of meaningful events.

Time can feel as though it unfolds like thread on a spindle[1]. Yet our memories do not strictly mirror this continuous flow of experience. Rather, everyday memories become woven into a tapestry of distinct and meaningful units, or episodes. This process of event segmentation helps divide the constant stream of sensory information into meaningful events, packaging experiences in ways that promote their durability and accessibility over time[2]. Because event segmentation is ubiquitous, relatively automatic, and foundational to healthy cognitive function[3,4], there has been intense interest in characterizing the factors that facilitate the discrete organization of episode memory.

The temporal stability of different types of contexts may play an important role in supporting this memory-chunking process. Increasing evidence shows that items encountered in a shared, or stable, context (e.g., appearing in the same room) become linked together in memory[5–10]. By contrast, when there is a sudden change in context (e.g., crossing through a doorway into a new room), items are thought to become separated in memory by a theoretical "event boundary" that defines a transition from one meaningful event to the next[11–13]. It is now well documented that fluctuations in the external world, such as space or perceptual features, influence how we encode and remember sequences of contextually-distinct events[5,11,13]. However, it is less clear how fluctuations in our *internal* world, such as emotional states, elicit similar effects on the episodic structure of memory. Decades of research show that emotional experiences become our most vivid and enduring memories[14–16]. By comparison, less work has examined how emotion shapes important temporal aspects of memory[17–19]. The aim of this study is to examine how shifting emotional states promote the organization of an otherwise continuous experience into distinct and memorable events.

How might fluctuating emotional states affect the organization of memory? One possibility is that emotions provide a strong *internal* context for linking and separating memories across time[20,21]. Initial support for this idea comes from evidence that more stable fluctuations in pupil-linked arousal levels relate to better encoding of temporal order memory and more compressed estimates of time between items – two behavioral indices of event integration[22]. By contrast, encountering an auditory boundary elicits a spike in arousal, as indexed by pupil dilation, that relates to subsequent impairments in temporal order memory and more expanded estimates of time between encoded items – two behavioral indices of event separation. Together, these findings suggest that continuity and change in arousal states influence the temporal structure of memory. Broadly, this is also

[1]University of California, Los Angeles, Department of Psychology, Los Angeles, CA, USA. [2]Columbia University, Department of Psychology, New York City, NY, USA. [3]Columbia University, Center for Science and Society, New York City, NY, USA. ✉e-mail: david.clewett@psych.ucla.edu

consistent with neurocognitive models of arousal – an elevated state of mental activation – and memory function. Moment-to-moment fluctuations in arousal systems signal the presence of salient information and promote encoding processes that etch these representations into memory more enduringly[23]. Given that arousal is also one of the primary dimensions of emotion[24], we propose that emotion dynamics might serve as another form of context that binds or separates adjacent episodes into contextually distinct memories.

Importantly, arousal is not the only defining feature of subjective emotional experience. Valence is also a key dimension of core affect[25,26], with negative and positive emotions often exerting different effects on the quality of episodic memory[27–29]. On one hand, negative emotional valence (i.e., how aversive something is) elicits strikingly similar effects on cognition as do event boundaries, including increased item-focused processing and focused attention[30,31]. This finding suggests that moments of strong negative emotion may disrupt sequential processing and enhance event segmentation effects in memory. On the other hand, positive emotions broaden the scope of attention[32–34] and enhance associative memory[27–29]. While these associative memory effects have been shown for static item-item associations, such as paired associates displayed side-by-side, it is unclear whether the mnemonic benefit of positive affective states also extends to temporal encoding processes. One compelling possibility is that positive emotions also boost relational processes *across time*, providing additional scaffolding for binding sequential associations in memory.

Accurately perceiving and remembering temporal duration and order is essential for supporting adaptive decision-making and learning[35–38], including for emotionally significant experiences. Until recently, however, emotion-memory research has focused primarily on how emotional stimuli influence recognition of items, their non-temporal features, or their probability of being retrieved in recall[21,39,40]. Additionally, attempts to investigate the role of emotion on temporal memory neglect the role of continuously shifting emotional states on memory for event structure within temporally extended sequences. Namely, most studies use individual emotional stimuli to elicit moments of emotional reactivity[41–43] as well as memory metrics that fail to capture how complex memories take form. For example, prior work has tested temporal source memory for information encountered during, but not surrounding, emotional events[44]. Other studies have tested temporal source judgments for longer-timescale events[18,42,45–47]. Further still, memory for the temporal order of image sequences or frames from naturalistic film clips are typically queried by presenting items one at a time[48,49] or all at once during retrieval[44]. Thus, there is a critical need to determine if memory for the structure of events is influenced by heterogenous emotional states. Here, we sought to address these limitations by investigating if fluctuations in music-induced emotions predict memory for the temporal structure and content of continuous item sequences.

Eliciting dynamic emotions through music has several key advantages over more traditional manipulations of emotional memory. Emotional images and words possess greater semantic relatedness with each other compared to non-emotional stimuli, thereby posing a challenge for dissociating emotion's distinct influence on memory[39,50]. Memory can also be biased by the amount of conceptual overlap between the thematic content of narratives and an arousal-inducing event[51]. Music, on the other hand, can reliably induce a range of emotions spanning the valence-arousal space with less explicit linguistic structure[52–54]. Unfamiliar pieces of instrumental music may help prevent individuals from relying on prior knowledge to scaffold episodic encoding processes[55]. Finally, different emotional reactions to music can be evoked by overlapping acoustic features, such as tone or tempo, enabling some level of control over the lower-level effects of perceptual change on memory encoding. Altogether, these features suggest that music is an effective method for studying the link between emotion dynamics and memory because it can reduce the influence of semantic relatedness, familiarity, and perceptual change on memory.

In the current study, we hired composers to write new musical compositions that evoke basic emotions, such as sadness and joy. During encoding, participants listened to different songs while studying sequences of neutral object images. We then queried event segmentation effects using both objective and subjective measures of temporal memory[6,22]. Critically, to track moment-to-moment changes in subjective emotional experience during encoding, we designed a novel emotional tracking tool, the *Emotion Compass*, that collects continuous ratings of emotional valence and arousal. Change-point analyses were used as a data-driven approach to identify structured patterns of change, or event boundaries, in these continuous emotion ratings. We then examined how both discrete emotional boundaries and continuous emotional change were related to memory for temporal and non-temporal aspects of episodic memory, including memory for order, subjective distance, individual items, and temporal source information.

In this work, we find that emotion dynamics mold otherwise neutral experiences into meaningful mnemonic events. Our results show that discrete and continuous changes in music-evoked emotions trigger event separation effects in memory, as indexed by subjective time dilation and impaired temporal memory for items spanning different emotional contexts. Furthermore, our results demonstrate that positive valence enhances associative encoding processes[28,29,33], such that reductions in highly negative affective states promote the binding and compression of memory, whereas shifts towards more negative emotional states promote event segmentation. Additionally, memories of item information that coincide with musical (i.e., perceptual) boundaries or valence boundaries are enhanced over the long term, particularly for items encountered during arousing and positive emotional states. With regard to temporal source memory, we find that all boundary types improve long-term memory for an item's original temporal context, with a strong benefit for items co-occurring with emotional context shifts that are highly arousing and positive. Overall these results show that emotion dynamics elicited by music shape the structure of long-term episodic memories, with positive affective states likely promoting the integration of information in memory across time and negative affective states promoting memory separation.

## Results

### Musical and emotional boundaries elicit event segmentation effects in memory

Our primary goal was to determine if meaningful shifts in emotional states function as event boundaries that segment memory (Fig. 1a). To this end, we first examined if discrete changes in musical and emotional contexts influenced memory for the subjective distance and temporal order between pairs of items from recently studied object lists (Fig. 1b, c). To create music with controlled perceptual and emotional context changes, we had professional musical composers write songs that conveyed four different basic emotional themes (joyous, anxious, sad, and calm). We then mixed the emotional themes into a series of unique 120 s compositions. Each composition contained three unique 30–40 s emotional segments that were connected by 6–9 s musical transition periods. These transitions helped reduce the likelihood that participants would perceive an obvious and abrupt perceptual change between segments (Supplementary Table 1 for the event structure design of each musical composition). Importantly, the compositions were designed to elicit different patterns of experienced valence and arousal to promote dynamic variability across emotional states. All compositions included no more than 2 emotional themes of the same valence (positive, negative) or arousal (high, low) across all 3 segments.

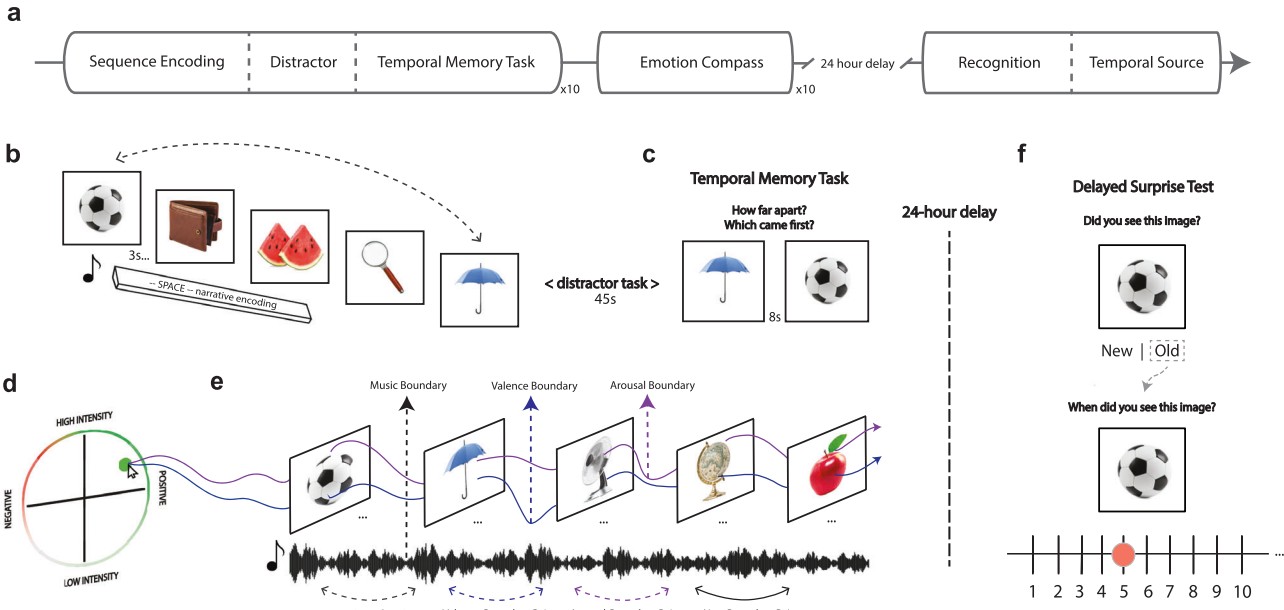

**Fig. 1 | Music emotional event segmentation task design. a** Timeline of experiment tasks. Participants performed a blocked item sequence encoding task while listening to different musical pieces. This task was followed by a continuous emotion-rating task in which participants re-listened to the music. Participants returned 24 h later to complete an item recognition and temporal source memory task. **b** Schematic of emotional event segmentation task. Participants encoded neutral object images while listening to musical pieces designed to elicit different basic emotions. **c** Temporal memory task. After encoding each musical sequence, participants' memory was probed for the temporal order and subjective temporal distance between different pairs of items from the prior sequence. **d** Emotion Compass task (see also Fig. 2 for more details). After the entire memory task was finished, participants rated their subjective moment-to-moment emotional reaction to the same musical pieces. **e** Emotion time-series mapping. Emotion Compass valence and arousal profiles were mapped back onto the timeline of item encoding. This enabled us to examine how structured changes in music-related emotion ratings influenced temporal and non-temporal aspects of encoding. **f** After a 24-h delay, participants returned and took a surprise item recognition and temporal source memory test for all items studied on Day 1. The object images in this figure are credited to: Aleksey Sagitov/Shutterstock.com; Bildagentur Zoonar GmbH/ Shutterstock.com; Anton Starikov/Shutterstock.com; Sergiy1975/Shutterstock.com; James.Pintar/Shutterstock.com; Preto Perola/Shutterstock.com; tanadtha lomakul/Shutterstock.com; and GalapagosPhoto/Shutterstock.com.

To identify musical, or auditory perceptual, event boundaries driven by acoustic changes in the music, we used a standard event segmentation approach[56–58] where a separate group of annotators ($n = 6$) listened to each song and pressed a button whenever they perceived a meaningful perceptual change in the music. A consensus perceptual boundary annotation was then constructed from these data by taking the average boundary time points across annotators. To track emotion dynamics induced by music, participants re-listened to all songs after the sequence encoding task and made continuous valence and arousal ratings using the *Emotion Compass* tool (Fig. 2a; for validation of the four normative emotions from participants' ratings, see Fig. 2c, d; also see Supplementary Figs. 1–3). We then used a data-driven approach to identify emotional valence and arousal boundaries in felt emotions during music listening. Specifically, we conducted change-point analyses on the valence and arousal time series for each song acquired from the *Emotion Compass*. The change-point algorithm identified when there was a significant change in the mean and slope of the valence and arousal rating timeseries, separately (Fig. 2b). The timestamps of these event boundaries were then aligned to the timeline of object presentations during the sequence encoding task. Object pairs that were to be tested for temporal order and distance memory were then labeled according to whether they spanned ("boundary-spanning" pair) or did not span ("non-boundary" pair) a perceptual, (i.e., musically-induced), valence, or arousal boundary (Fig. 1e). As illustrated in the arousal and valence ratings in Supplementary Fig. 5, the change-point algorithm appropriately labeled the different valence and arousal boundary types.

We first examined how different event boundaries influence subjective representations of time in memory. We found that both perceptual boundaries ($\beta = 0.02$, standard error (SE) = 0.008, $\chi 2$ (1) = 7.28, $p = 0.007$, 95% CI = [0.01, 0.04]) and valence boundaries

($\beta = 0.034$, standard error (SE) = 0.008, $\chi 2$ (1) = 16.8, $p < 0.001$, 95% CI = [0.02, 0.05]) elicited a subjective time dilation effect in memory, such that boundary-spanning pairs were judged as having appeared farther apart in time than non-boundary pairs during encoding, despite being the same objective distance apart (Fig. 3a). By contrast, there was no statistically significant effect of arousal-related boundaries on subjective time dilation ($\beta = -0.002$, standard error (SE) = 0.008, $\chi 2$ (1) = 0.084, $p = 0.772$, 95% CI = [−0.01, 0.02]). Post-hoc contrasts of the estimated marginal means show a statistically significant subjective dilation for item pairs spanning perceptual boundaries (temporal distance difference = 0.045, z = 2.7, $p = 0.007$, 95% CI of the difference = [0.01, 0.07]) as well as valence boundaries (distance difference = 0.066, z = 3.938, $p < 0.001$, 95% CI of the difference = [0.04, 0.1]), but not arousal boundaries (temporal distance difference = 0.005, z = −0.289, $p = 0.772$, 95% CI of the difference = [−0.04, 0.03]).

Next, we examined if event boundaries disrupt temporal binding processes to elicit memory separation. We found similar segmentation effects of event boundaries on objective aspects of temporal memory. Specifically, relative to non-boundary pairs, temporal order memory was statistically significantly impaired for item pairs spanning perceptual boundaries ($\beta = -0.09$, standard error (SE) = 0.027, $\chi 2$ (1) = 13.86, $p < 0.001$, 95% CI = [−0.14, −0.03]) and valence boundaries ($\beta = -0.15$, standard error (SE) = 0.027, $\chi 2$ (1) = 32.78, $p < 0.001$, 95% CI = [−0.21, −0.1]), but not arousal boundaries ($\beta < 0.001$, standard error (SE) = 0.027, $\chi 2$ (1) < 0.001, $p = 0.992$, 95% CI = [−0.05, 0.05]; Fig. 3b). Post-hoc contrasts of the estimated marginal means showed a statistically significant temporal order memory impairment for item pairs spanning perceptual boundaries (temporal order difference = 0.201, z = 3.733, $p < 0.001$, 95% CI of the difference = [0.07, 0.28]), as well as valence boundaries (temporal order difference = 0.31, z = 5.739,

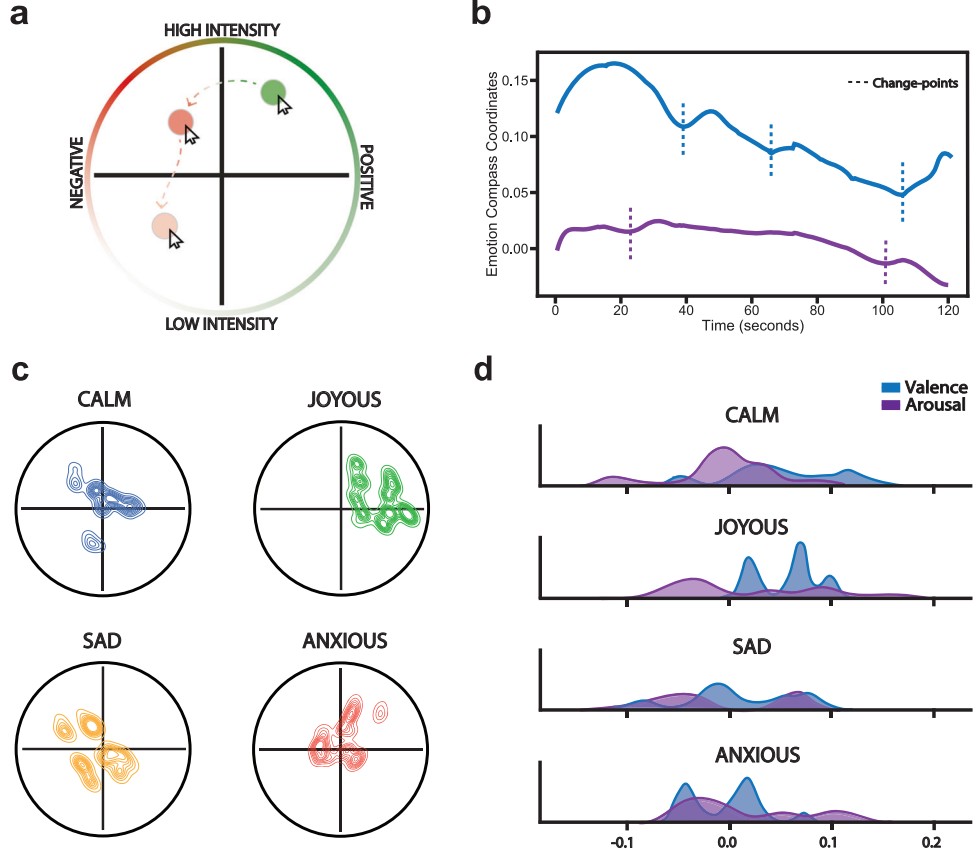

**Fig. 2 | Emotion Compass mechanics and emotion category profiles.**
**a** Schematic of the Emotion Compass. After performing encoding and temporal memory tasks, all participants re-listened to each song while dynamically rating their valence and arousal in real-time (sampled at 6 Hz). A planchette (green and red circles) tracked participants' cursor position and changed color depending on the placement within the compass. **b** Average emotional valence (blue) and arousal (purple) temporal profiles of an example song (joy-anxious-sad). Emotional valence and arousal ratings were averaged across all participants and mapped back onto the timeseries of object encoding. Data-driven valence and arousal 'event boundaries' (dotted lines), or drastic shifts in emotion, were also identified using change-point analyses. **c** Two-dimensional kernel density plots of each emotion category (Calm, Anxious, Sad, and Joyous) across all participants and songs. **d** One-dimensional kernel density plots for average music-induced valence and arousal ratings by emotion category of the song. Source data for **b, c,** and **d** are provided as Source Data File.

---

$p < 0.001$, 95% CI of the difference = [0.2, 0.42]), but not arousal boundaries (temporal order difference = −0.001, z = 0.014, $p = 0.992$, 95% CI of the difference = [−0.1, 0.1]). Taken together, these change-point results suggest that large shifts in emotional valence create event boundaries that segment episodic memory.

One potential issue with this music manipulation is that the perceptual features of the music itself (i.e., acoustics) cannot be completely separated from the emotions they elicit. Thus, we wanted to verify that the effects of emotional boundaries on temporal memory were not simply an artifact of perceptual change. A qualitative comparison of the boundary labels for each pair demonstrates that there were clear differences between the time points of boundaries, despite some expected overlap (Supplementary Figs. 4, 5).

We next performed hierarchical linear modeling to test if valence boundaries account for memory segmentation effects above and beyond the effects of perceptual boundaries. The results revealed that both subjective dilation ($\beta = 0.033$, standard error (SE) = 0.008, $\chi2$ (1) = 15.5, $p < 0.001$, 95% CI = [0.016, 0.05]) and temporal order impairment ($\beta = -0.15$, standard error (SE) = 0.027, $\chi2$ (1) = 30.24, $p < 0.001$, 95% CI = [−0.2, −0.1]) were statistically significantly related to valence boundaries even when controlling for the effects of perceptual boundaries. These findings suggest that emotional change, specifically in experienced positive affect evoked by the music, elicits memory separation effects beyond mere perceptual change.

## Continuous change in subjective emotion influences the separation and integration of episodic memories

The results so far demonstrate that discrete musical and valence boundaries relate to the separation of adjacent events in memory. However, emotions also evolve both gradually and subtly over time. Event segmentation effects in memory might thereby occur at multiple timescales and levels of emotional change. These potentially meaningful shifts in felt emotions might not be detected using more discrete change-point-defined event boundaries, which required constraining boundary labels to approximately two time points in the songs. To test these possibilities, we used an alternative approach of extracting both absolute and signed changes in subjective valence and arousal ratings from the *Emotion Compass* between every neutral to-be-tested item pair (Fig. 4a). One key advantage of this approach is that it obviates the need to pre-define and statistically limit the number of event boundaries that could be encountered during a given item sequence. As before, we used linear mixed modeling to relate these continuous item pair-level measures of emotional states to their corresponding subjective temporal distance and temporal memory outcomes.

Consistent with the discrete boundary results with change-points, we found that the absolute difference in felt emotional valence between otherwise neutral object pairs during encoding was statistically significantly positively related to the magnitude of subjective time dilation between those same item pairs ($\beta = 0.019$, standard error (SE) = 0.009, $\chi2$ (1) = 5, $p = 0.025$, 95% CI = [0.002, 0.036]; Fig. 4b, top).

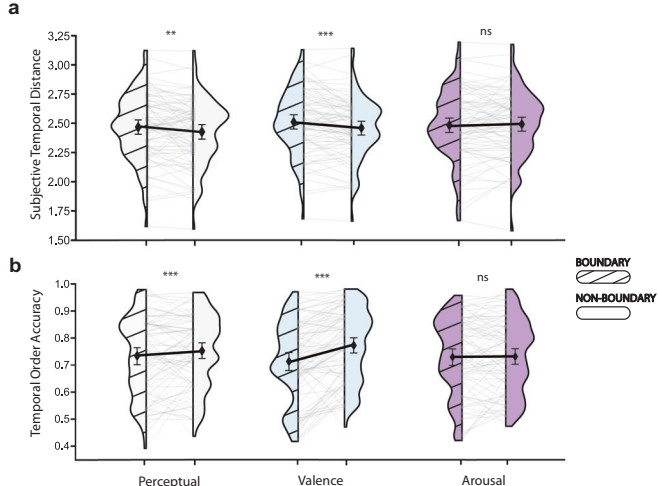

**Fig. 3 | Musical and valence boundaries influence the temporal structure of memory. a** Wing-plots showing subjective time dilation in memory for item pairs spanning musical (grey, left) and valence (blue, middle), but not arousal (purple, right) boundaries compared to item pairs that did not span boundaries. **b** Wing-plots showing impaired temporal order for item pairs spanning a musical (perceptual) event boundary (grey, left) and valence event boundary (blue, middle), but not an arousal event boundary (purple, right) compared to item pairs that did not span boundaries. Wing plots represent the distribution of participant datapoints, and connected gray lines represent the effects for individual participants ($N = 81$). Block diamonds represent mean memory performance across participants. Error bars = 95% bootstrapped CI. Linear mixed effects models indicated that perceptual (i.e., musical) and valence boundaries compared to non-boundaries were associated with subjective time dilation (perceptual: $p = 0.007$; valence: $p < 0.001$) and impaired temporal order memory (perceptual: $p < 0.001$; valence: $p < 0.001$). **\*\***$p < 0.01$, **\*\*\***$p < 0.001$. Source data are provided as Source Data File.

Larger absolute differences in valence between to-be-tested item pairs were also statistically significantly related to greater impairments in temporal order memory for their corresponding pair of items ($\beta = -0.08$, standard error (SE) = 0.028, $\chi2$ (1) = 7.23, $p = 0.007$, 95% CI = [−0.13, −0.02]; Fig. 4c, top). These results suggest that the magnitude of emotional shifts between two items during the ongoing experience can amplify their separation in later memory.

Interestingly, the signed change in valence between item pairs was statistically significantly related to the degree of subjective temporal compression in memory, such that as felt valence that accompanied object pairs increased (i.e., became more positive), they were more likely to be judged as being studied closer together in time ($\beta = -0.022$, standard error (SE) = 0.009, $\chi2$ (1) = 6.61, $p = 0.013$, 95% CI = [−0.038, −0.005]; Fig. 4b, **bottom**). A similar memory integration effect was observed for temporal order memory, with increasingly positive valence changes being statistically significantly related to better temporal order memory for those item pairs ($\beta = 0.076$, standard error (SE) = 0.037, $\chi2$ (1) = 4.19, $p = 0.041$, 95% CI = [0.003, 0.15]; Fig. 4c, **bottom**). Thus, the direction of change for subjectively experienced valence during music listening had divergent effects on memory integration processes.

Because this was a linear effect, it is somewhat unclear if integration was driven by emotions moving from a more negative state to a less negatively-valenced state (staying within the left hemisphere of the Emotion Compass and drifting toward the y-axis; see Fig. 4d) or moving from a less positive state to more positively-valenced state (staying within the right hemisphere by drifting away from the y-axis). To determine the nature of these signed valence effects, we conducted separate non-preregistered exploratory regressions for item pairs encoded during purely positive versus purely negative states (i.e., both items were associated with negative or positive emotion). We found

that under negative states, moving from a more negative state to a less negatively-valenced state led to statistically significantly more compressed estimates of time ($\beta = -0.047$, standard error (SE) = 0.01, $\chi2$ (1) = 7.36, $p = 0.006$, 95% CI = [−0.08, −0.02]; Fig. 4e, top panel) as well as enhanced temporal order memory ($\beta = 0.12$, standard error (SE) = 0.04, $\chi2$ (1) = 5.36, $p = 0.008$, 95% CI = [0.03, 0.24]; Fig. 4f, top panel). By contrast, moving from a less positively valenced state to a more positively-valenced state was not statistically significantly related to either subjective time ($\beta = -0.013$, standard error (SE) = 0.009, $\chi2$ (1) = 1.41, $p = 0.160$, 95% CI = [−0.04, 0.01]; Fig. 4e, **bottom panel**) or temporal order memory ($\beta = -0.01$, standard error (SE) = 0.003, $\chi2$ (1) = 1.3, $p = 0.933$, 95% CI = [−0.09, 0.09]; Fig. 4f, **bottom panel**). These findings suggest that event integration in memory is primarily driven by a shift away from highly a negative emotional state to a less negative emotional state (i.e., a reduction in negative valence specifically).

Unlike the robust effects of valence on temporal memory, neither absolute change nor signed change in arousal ratings were statistically significantly related to temporal distance ratings (absolute change: $\beta = 0.006$, standard error (SE) = 0.008, $\chi2$ (1) = 0.437, $p = 0.508$, 95% CI = [−0.01, 0.02]; signed change: $\beta = 0.002$, standard error (SE) = 0.009, $\chi2$ (1) = 0.029, $p = 0.862$, 95% CI = [−0.02, 0.02]) or temporal order memory (absolute change: $\beta = -0.01$, standard error (SE) = 0.06, $\chi2$ (1) = 0.029, $p = 0.869$, 95% CI = [−0.06, 0.05]; signed change: $\beta = 0.036$, standard error (SE) = 0.032, $\chi2$ (1) = 1.28, $p = 0.258$, 95% CI = [−0.03, 0.1]).

Altogether, these results demonstrate that experiencing a significant shift in emotional valence, irrespective of its directionality, elicits event segmentation effects in memory. However, the direction of this valence shift had divergent effects on the temporal structure of memory. When this emotional shift occurred in a positive direction, participants were more likely to bind and compress sequential representations in memory. By contrast, when this shift occurred in a negative direction, participants were more likely to segment temporally adjacent experiences in memory.

## Long-term memory is enhanced for information encoded at both musical and emotional event boundaries

Leading models of event segmentation emphasize the importance of retaining information encountered at event boundaries for strengthening long-term memories[12,59]. Indeed, this idea is supported by empirical evidence showing that perceptual boundaries enhance free recall[60], item recognition[59], and source memory[22,61,62] for concurrent information. Thus, in a non-preregistered exploratory analysis, we next examined if long-term memory for objects and their temporal contexts would be strengthened if they were encountered during a musical and/or emotional change (see Fig. 5a).

A logistic mixed effects regression revealed a statistically significant main effect of Boundary Type on item memory ($\beta = 0.064$, standard error (SE) = 0.018, $\chi2$ (1) = 13.1, $p < 0.001$, 95% CI = [0.03, 0.1]). Post-hoc pairwise contrasts with FDR corrections revealed a statistically significant enhancement of recognition for items that were encountered concurrently with both perceptual boundaries (recognition difference =, $z = 4.11$, $P_{FDR} < 0.001$, 95% CI of the difference = [0.1, 0.42]) and valence boundaries (recognition difference =, $z = 2.996$, $P_{FDR} = 0.008$, 95% CI of the difference = [0.03, 0.35]) compared to items not encountered during boundaries. This item enhancement effect was not statistically significant for arousal-related boundaries ($z = 1.23$, $P_{FDR} = 0.261$, 95% CI of the difference = [−0.08, 0.22]).

In addition to modulating item encoding, we also predicted that event boundaries would enhance *temporal source* binding in long-term memory, given the known effects of boundaries on enhancing source memory for items and other forms of context (e.g., colored border or accompanying sounds). Temporal displacement, or the amount of error in temporal source memory, was determined by calculating the

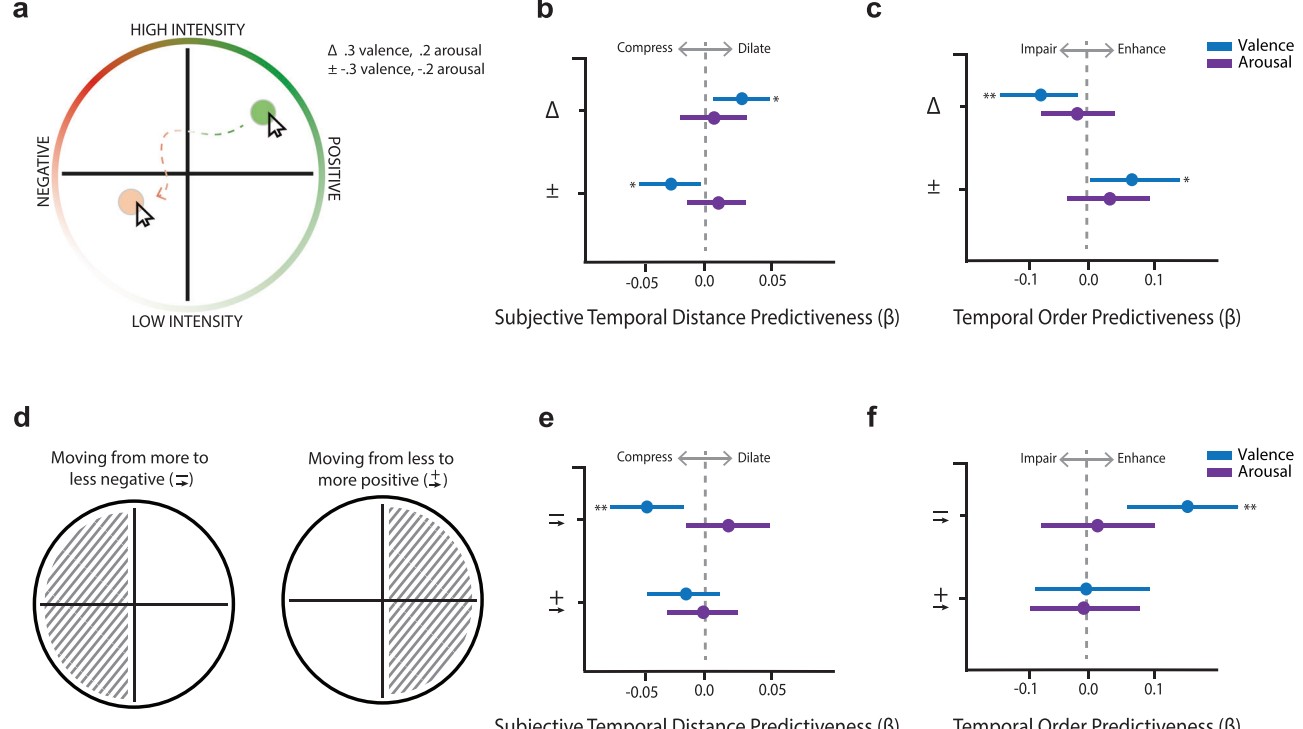

**Fig. 4 | Continuous emotional fluctuations predict subjective and objective aspects of temporal memory. a** Two metrics of shifting emotional states were extracted to examine how changes in experienced valence and arousal relate to the temporal binding between corresponding item pairs in each sequence: 1) absolute difference in emotion ratings, and 2) signed change in emotion ratings. Linear mixed models indicated that: **b** Absolute change in valence positively relates to subjective temporal dilation (top; $p = 0.025$), whereas increasing valence relates to subjective temporal compression in memory (bottom; $p = 0.013$). **c** Absolute change in valence positively relates to temporal order impairment (top; $p = 0.007$), whereas increasing valence relates to temporal order improvement (bottom; $p = 0.041$). **d** Signed change in valence was then broken down by whether coordinates moved from a more to less negative state (i.e., within the left hemisphere of

the Emotion Compass) or moved from a less to more positive state (i.e., stayed within the right hemisphere of the Emotion Compass). Linear mixed models indicated that: **e** Moving from a more negative state to a less negative state was related to temporal compression (top; $p = 0.008$). This relationship between valence shifts and temporal distance ratings was not found when participants' emotions shifted from a less positive state to a more positive state (bottom; $p = 0.160$). **f** Moving from a more negative state to a less negative state was related to better temporal order memory (top; $p = 0.008$). However, moving from a less positive state to a more positive state was not related to better temporal order memory (bottom; $p = 0.933$). In **b**, **c**, **e**, and **f**, dots represent linear mixed model betas. $N = 81$ participants for all models. Error bars = 95% bootstrapped CI. *$p < 0.05$, **$p < 0.01$. Source data for **b**, **c**, **e**, and **f** are provided as Source Data File.

absolute difference between an item's true position in its list and the list position that was endorsed by participants during the memory test (see Fig. 5a). A linear mixed effects regression revealed a statistically significant main effect of Boundary Type on temporal source memory for concurrently presented items ($\beta = -0.11$, standard error (SE) = 0.008, $\chi2$ (1) = 192.77, $p < 0.001$, 95% CI = [−0.13, −0.09]). Post-hoc pairwise contrasts with FDR corrections revealed statistically significant lower temporal source errors for items that were encountered during perceptual (z = −5.83, $P_{FDR} < 0.001$, 95% CI of the difference = [−1.1, −0.42]), valence (z = −10.94, $P_{FDR} < 0.001$, 95% CI of the difference = [−1.7, −1.1]), and arousal (z = −9.08, $P_{FDR} < 0.001$, 95% CI of the difference = [−1.4, −0.8]) boundaries compared to items not encountered during boundaries. Further, post-hoc contrasts revealed statistically significantly lower temporal source displacement (i.e., more accurate responses) for items encountered during valence (z = −3.81, $P_{FDR} < 0.001$, 95% CI of the difference = [−1.1, −0.21]) and arousal (z = −2.2, $P_{FDR} = 0.031$, 95% CI of the difference = [−0.8, −0.06]) boundaries compared to music boundaries. Temporal source memory did not statistically significantly differ between items encountered during valence and arousal boundaries (z = 1.7, $P_{FDR} = 0.089$).

Because the perceptual boundaries – and by extension, emotional boundaries – were encountered approximately 1/3$^{rd}$ and 2/3$^{rds}$ into a given list, it is possible that boundary-related enhancements in temporal source memory were simply driven by list position effects (Supplemental Fig. 6). To account for this potential confound, we

investigated if temporal source memory effects were localized to the middle of the lists. While temporal source displacement decreased (i.e., temporal source memory improved) as a function of an item's proximity to the middle of the list (Supplemental Fig. 7), our effects of interest – namely that the timing of boundary items were remembered better than non-boundary items – were specific to the beginning and the end of the lists, not the middle (Supplemental Fig. 9).

Together these findings suggest that musical and emotional boundaries not only enhance the memorability of concurrent information but also improve memory for its temporal context. Further, the temporal context of emotional boundaries may be more memorable than perceptual boundaries alone, suggesting that emotion is particularly effective at enhancing item and associative representations in long-term memory.

### High-arousal positive emotions anchor item representations in long-term memory

The previous delayed memory analyses revealed that emotional and musical event boundaries enhance long-term item recognition and temporal source memory. Beyond discrete, dichotomous effects of event boundaries on memory, however, emotional states could also color the processing and encoding of individual items more subtly; that is, the emotions experienced while viewing a neutral image can imbue it with new meaning[63]. Thus, the current state of emotion at encoding – as opposed to whether overall emotional states are stable

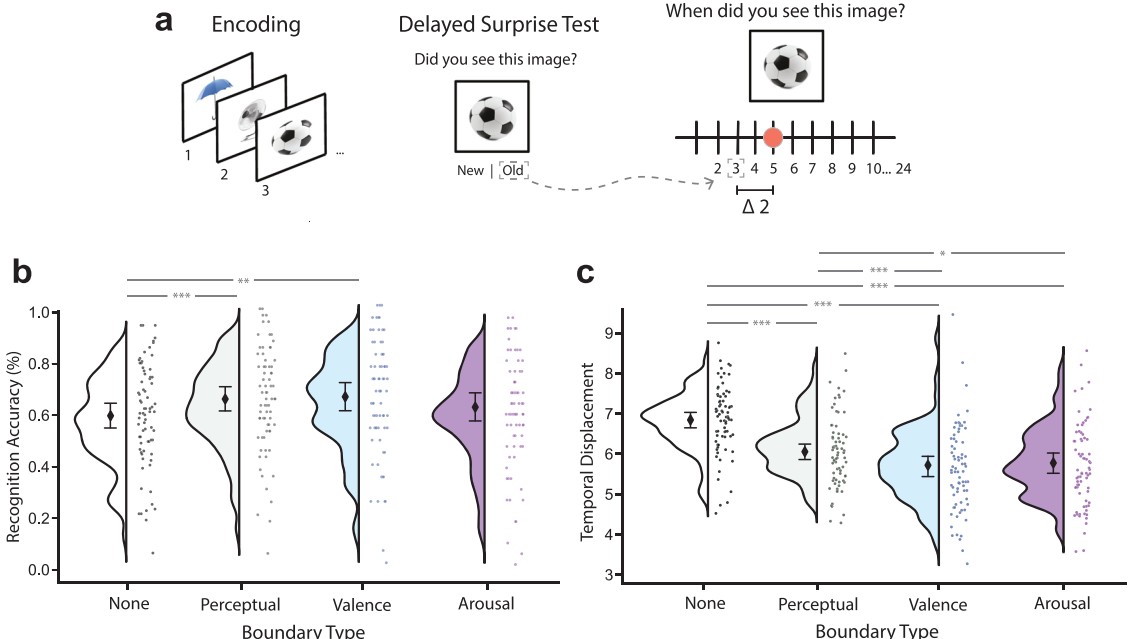

**Fig. 5 | Musical and emotional event boundaries enhance long-term memory for objects and their temporal contexts. a** Participants returned 24 h after the sequence encoding task to take a surprise recognition and temporal source memory task. On each trial of the memory test, participants first indicated if an object image was "Old" or "New". If they endorsed the item as "Old", they were then queried for temporal source memory. In this source test, participants placed a dot on a slider indicating when the object had appeared within its respective 24-item list. Temporal source memory accuracy was measured by temporal displacement, which is defined as the absolute difference between the endorsed item position on the slider (out of 24 positions) and the original item position. **b** Linear mixed models indicate that after a 24-h delay, neutral objects encoded during perceptual

($p < 0.001$)and emotional valence ($p < 0.001$) boundaries were better remembered compared to items not encoded at boundaries. **c** Linear mixed models also indicate that displacement in temporal source, or temporal context, memory was lower for objects encoded during perceptual ($p < 0.001$), arousal ($p = 0.031$), and valence ($p < 0.001$) event boundaries compared to items not encoded at boundaries. Black diamonds represent the mean across all participants ($N = 73$). Error bars = 95% bootstrapped CI. *p < 0.05, **p < 0.01, ***p < 0.001. All $p$s are FDR-corrected across conditions. Source data for b and c are provided as Source Data File. The object images in this figure are credited to: Aleksey Sagitov/Shutterstock.com; Bilda-gentur Zoonar GmbH/Shutterstock.com; and Anton Starikov/Shutterstock.com.

or changing, per se – may also have a strong influence over memory. Indeed, much work shows that emotional contexts, such as faces or scenes, can modulate memory for concurrent neutral items[64,65].

To examine these modulatory effects of felt emotions on item processing, we next examined how music-induced emotional states relate to the delayed item and associative memory for the neutral object images. A logistic mixed effects regression revealed a statistically significant valence-by-arousal interaction effect on delayed item recognition ($\beta = 0.05$, standard error (SE) = 0.23, $\chi2$ (3) = 8.73, $p = 0.033$, 95% CI = [0.05, 0.01]). To disentangle this interaction effect, we performed follow-up simple slopes analyses to examine how arousal influences item memory at different levels of valence. As shown in Fig. 6a, b, we found that neutral items encoded during high-valence music (i.e., more positive emotions) were statistically significantly better remembered as a function of increasing arousal (1 SD above the mean = 0.073; $\beta = 0.08$, $p < 0.001$, 95% CI = [0.03, 0.14]). By contrast, there was no statistically significant association between arousal and recognition success when the music was rated as being more negative (1 SD below the mean = −0.021; $\beta = −0.01$, $p = 0.721$, 95% CI = [−0.01, 0.04]).

**High-arousal positive emotions enhance temporal source binding in long-term memory**
Following this same logic, we next examined if felt emotions influenced memory for an item's temporal context, or position within its 24-item list. A mixed effects regression revealed a statistically significant valence-by-arousal interaction effect on temporal source memory ($\beta = −0.711$, standard error (SE) = 0.072, $\chi2$ (3) = 8.73, $p < 0.001$, 95% CI = [−0.85, −0.56]). Here, temporal source memory accuracy was measured by temporal displacement, which is defined

as the absolute difference between the endorsed item position on the slider (out of 24 possible positions) and the original item position in a list. Thus, lower temporal displacement values are indicative of less error, or better temporal source memory. Follow-up simple slopes analyses revealed that the temporal index of items encoded during high-valence music (i.e., more positive emotion) was statistically significantly better remembered as a function of increasing arousal (high = 0.073, or 1 SD above the mean; $\beta = −0.33$, $p < 0.001$, 95% CI = [−0.44, −0.21]). Conversely, memory for the temporal source of items encoded during low-valence (i.e., more negative emotion) music was statistically significantly impaired as a function of increasing arousal (low = −0.021, or 1 SD below the mean; $\beta = 0.13$, $p < 0.001$, 95% CI = [0.03, 0.24]; see Fig. 6c, d).

Taken together, our Day 2 findings suggest that music-induced emotional states influence long-term memory representations for neutral items and their temporal contexts. Importantly, these effects also differed as a function of the subjective arousal and valence experienced by music listeners. On one hand, highly arousing, positively valenced emotional states elicited by music enhanced both item (i.e., object recognition) and temporal source memory (i.e., lower temporal displacement in list position). On the other hand, highly arousing, negatively-valenced emotional states impaired temporal source memory (i.e., greater temporal displacement) but had no statistically significant effect on delayed item recognition.

## Discussion
Our goal in the present study was to determine if dynamic fluctuations in emotional states facilitate encoding of discrete events in memory. While it has been shown that changes in external contexts, such as walking through a doorway, lead to the separation of events in

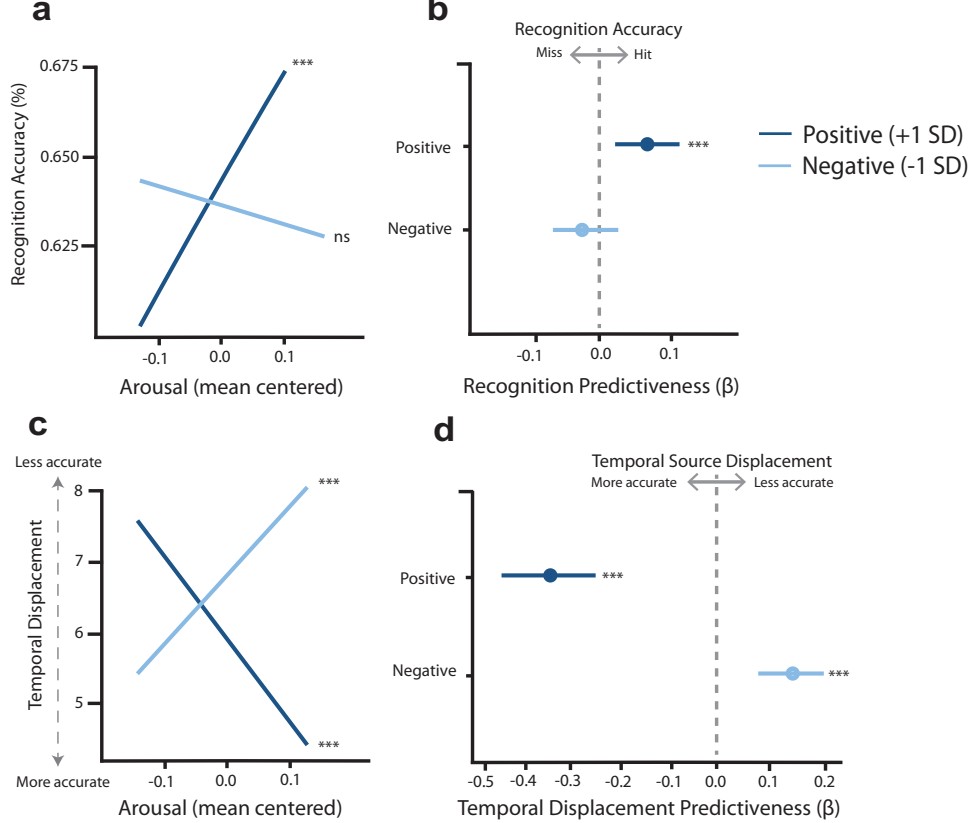

**Fig. 6 | Subjective valence and arousal during music listening relates to delayed object recognition and temporal source memory. a** Simple slopes analysis. Linear mixed models indicate that emotional valence and arousal elicited by music interacted to affect recognition memory, whereby positive (high-valence) items were better remembered as a function of increasing arousal ($p < 0.001$). **b** Beta values and their error extracted from simple slope regressions predicting recognition. **c** Linear mixed models indicate that subjective valence and arousal elicited by music interacted to affect temporal displacement in memory, whereby positive (high-valence) items' temporal positions were remembered better as a

function of increasing arousal ($p < 0.001$). Memory for temporal positions was impaired if those objects co-occurred during negative (low-valence) moments as a function of increasing arousal ($p < 0.001$). **d** Beta values and their error extracted from simple slope regressions predicting temporal displacement. Dark blue lines = valence 1 SD above the mean valence for all songs; light blue lines = valence 1 SD below the mean valence for all songs. In **c**, and **d**, dots represent linear mixed model betas. $N = 73$ participants for all models. Error bars = 95% bootstrapped CI. ***$p < 0.001$. Source data are provided as Source Data File.

memory, it is less clear if changes in emotional states also shape the discrete, episodic structure of memory. To investigate this question, we hired composers to write new music that induced a range of emotional states, while also controlling certain acoustic and temporal features. We then developed a novel emotion-tracking tool, the *Emotion Compass*, that is sensitive to fine-scaled temporal fluctuations in subjective emotional experience. Finally, we combined these techniques with a musical event segmentation paradigm to investigate how emotion dynamics influence memory for the temporal unfolding of events.

Our findings reveal that fluctuating emotional states may function as a strong feature of an internal context that links and separates memories across time. Namely, the temporal stability of information from our internal world helps shapes episodic memory like external contextual cues, such as stability and change in the surrounding spatial context[4,5,18]. We found that significant changes in the perceptual features of music, as well as emotional valence, predicted subjective time dilation effects in memory as well as impaired temporal order memory – two behavioral indices of event segmentation[7]. Critically, these drastic shifts in emotional valence during encoding explained variability in mnemonic segmentation effects above and beyond the lower-level effects of musical change. Emotional valence might therefore provide a unique scaffolding for encoding and organizing memories of novel emotional episodes. Further, we found that neutral items and their temporal source information encoded at emotional event

boundaries were remembered better after a 24-h delay compared to within-event items, suggesting that moments of meaningful emotional change also influence processing of non-temporal aspects of episodic memory. Moreover, experiencing high-positive, high-arousal emotions during music listening facilitated the encoding of otherwise neutral items and their temporal contexts in long-term memory. Together these findings reveal that emotion dynamics shape the temporal organization of events in memory, with the arousal and valence of felt emotions exerting complex effects on both the initial and long-term structuring of episodic memory. Our data also reveal divergent effects of emotional valence on memory organization processes.

One of our critical findings was that the degree and quality (i.e., direction and valence) of emotional change influence the separation or binding of sequential representations in memory. Experiencing a shift towards more positive emotional states led to greater temporal memory integration, as indexed by more compressed estimates of subjective temporal distance and better temporal order memory. This finding, along with our results showing that neutral images encoded in a high-valence, high-arousing state were best remembered for their representation (recognition) and timing (temporal source) after a 24-h delay, suggest a memory-enhancing role of positively valenced states. The music itself also serves as a highly rewarding stimulus, which may enhance encoding and retrieval of items encountered alongside positive affect[66]. For instance, other work shows that positively

valenced musical pieces[67,68] as well as accompanying lyrics[69] are better recognized than negative pieces after a delay.

Interestingly, the valence-dependent shifts we observed on temporal memory appeared to be specific to the domain of negative affect. We found that experiencing a shift away from highly negative to less negatively valenced emotional states, rather than from less positive to more positively valence states, enhanced event integration processes in memory[32]. Conversely, this relationship can also be interpreted as a drastic shift towards negative states resulting in stronger event segmentation effects in memory. Indeed, this finding accords with evidence that negative stimuli tend to narrow the focus of attention on individual pieces of information, which in turn impairs relational strategies that support encoding temporal associations[70]. It also aligns with research showing that more item-focused encoding under neutral conditions disrupts temporal binding processes for temporally extended sequences[71] as well as work showing that negative emotions disrupt binding and retrieval of temporal context information[17,20,21]. Importantly, this valence-specific effect suggests that alleviating negative states rather than simply adding more positive valence to an already positive state helps preserve memories of the order and structure of distinct events. While the clinical relevance of our findings has yet to be tested, our results raise the possibility that improving the coherence of memory may have therapeutic benefits in disorders characterized by intense negative affect and memory disorganization. Future translational research should test if positive affect inductions improve memory for the order, timing, and causal relationships between meaningful events for individuals who suffer from highly negative emotions.

Why would increases in positive emotional states enhance temporal encoding processes and both item and source features of memory? The broaden-and-build theory of positive emotion posits that positive affect increases flexible cognitive processing and leads to enhanced integration of multi-featural information[33,34]. Furthermore, a growing body of literature has linked positive affect to enhanced associative memory[28], possibly via behavioral activation[29,72–74], or a state of increased motivation for exploration and novelty-seeking. It has been recently proposed that behavioral activation promotes the integration of contextual details in memory by shifting cognitive resources toward the representation of higher-order features of events, such as an object's background and surrounding items[29]. This idea dovetails with earlier work proposing that positive affect promotes the use of relational processing more broadly[7,75]. Therefore, musically-induced increases in valence may promote a state of flexible and relational cognitive processing, resulting in deeper encoding of temporal context information and stronger mnemonic binding processes.

Inducing positive moods through music has also been shown to benefit different types of memory, including boosting autobiographical memory retrieval in Alzheimer's patients[76], recognition of abstract images[77], as well as both visuospatial and navigational working memory[78]. Studies investigating mood-dependent learning show that encoding and retrieving information in similar emotional contexts elicited by music facilitates memory performance[79,80]. Prior work on moods more broadly is limited, however, in that it focuses on the effects of a sustained and singular affective state following mood induction. That is, the success of mood inductions is typically assessed with a single pre-to-post induction rating, and it is assumed that these mood changes affect subsequent encoding processes uniformly. However, this assumption fails to capture the temporally dynamic nature of emotion and overlooks the possibility that moment-to-moment emotion fluctuations are meaningfully related to variability in encoding processes across time. Indeed, unlike previous work using pre-encoding mood induction manipulations or static ratings of affect, we show that finer-scaled and temporally dynamic fluctuations in music-evoked emotions affect the structure and organization of

memory. The ability to track continuous emotional states thereby opens many avenues for exploring how emotion influences encoding processes as they unfold.

Although it has long been hypothesized that music may provide a salient encoding context for binding information in episodic memory[66,81], results have been mixed with regard to the potential benefits of background music on episodic memory. Prior studies have shown that unfamiliar, non-lyrical background music can aid in source memory for verbal information. Whereas familiar music may be distracting and hinder memory[82], unfamiliar music can provide new contexts that promote memory binding and subsequent retrieval[66]. This beneficial effect of music on verbal episodic memory appears to be even stronger when the music is more rewarding, possibly due to interactions between reward-related brain networks and brain regions that support episodic memory[83]. Background music has also been shown to elicit chunking effects in free recall, suggesting that music enhances inter-item associative binding[66,84]. Here, we extend previous findings by showing that the memory-enhancing effects of music are not limited to the verbal domain[66].

In addition to influencing the temporal structure of memory, we found that emotional and perceptual boundaries led to enhanced delayed recognition and temporal source memory for concurrent neutral items. Previous research has argued for a meaningful link between event boundaries and memorability[59,85–87]. For instance, leading theories of event cognition posit that boundaries provide opportunities for updating active representations of the current environment, or 'event model', prioritizing the processing of item and source details that define a novel event[2,11,59]. This organizational process also facilitates retrieval and comprehension of distinct episodic events by limiting competitive retrieval[88]. Supporting these ideas, the presence of different types of boundaries, such as doorways[88] or causal breaks in stories, can enhance the memorability of wordlists as well as memory for and comprehension of short linguistic narratives[89]. Further, work on prosodic segmentation in music has shown that active segmentation promotes memory encoding and comprehension of musical scores[81]. It is noteworthy that while it has long been thought that event boundaries enhance item recognition, these effects were observed only five seconds after an event boundary[59]. Thus, the current results complement and extend existing findings by demonstrating long-term consequences of different types of context shifts on memory for concurrent item information. Alongside item memory enhancements, we found that delayed temporal source memory was better for neutral items encountered at valence and arousal boundaries compared to neutral items encountered at musical (i.e., perceptual) boundaries or no boundaries. These results suggest that emotional boundaries may create temporal landmarks in memory that aid with remembering 'when' something occurred. One interesting question for future research is whether these item and source memory effects from a highly controlled manipulation generalize to memory for more complex emotional narratives. Event boundaries not only function to segment experience but are also the most informative moments of a narrative's semantic structure and overall meaning[10,90]. Information encountered within a stable context is contiguous and redundant, making it less important to encode and integrate into a larger narrative. Our findings reveal that emotional moments are strong boundaries in unfolding experiences, perhaps signaling the behaviorally relevant moments most needed for remembering and comprehending complex events.

One possible limitation of the present study is that neither continuous nor discrete (i.e., change-point) changes in emotional arousal were related to immediate tests of temporal memory. We observed comparable variability in both valence and arousal ratings across music listening and found that items that span valence and arousal change points coincide with prominent changes in valence and arousal, respectively. Participants also made emotion ratings consistent

with the basic emotion categories of the songs designed by the composers. For these reasons, it is unlikely that a lack of arousal effects was driven by participants being unable to simultaneously track changes in both valence and arousal during music listening. However, because we collapsed emotional ratings across participants to account for variability in online cursor tracking and to retain otherwise well-performing participants for analysis, there is a possibility that the song-level arousal ratings were not sensitive enough to account for variance in temporal order and temporal distance memory.

It could be the case that individuals primarily rely on valence information as an effective encoding strategy for committing sequential items to memory. However, high-valence, high-arousal states were related to enhanced delayed recognition and temporal source memory for otherwise neutral items. The temporal source of items encountered at arousal-related change points was also better remembered than the temporal source of items encountered at perceptual boundaries or non-boundaries, suggesting that large shifts in emotional arousal were still important for enhancing individual item representations and their surrounding temporal details in long-term memory.

A final possibility is that while changes in valence may promote segmentation effects in the immediate retrieval of temporal memory, the effects of emotional arousal may require a longer period of consolidation to emerge. Indeed, the effects of emotional arousal on memory tend to become more apparent after a delay[91]. While this consolidation prediction is beyond the scope of the current study, we recommend that future research investigate if valence and arousal have unique influences on event segmentation and temporal memory more generally, and whether certain dimensions of emotion (i.e., valence) are more meaningful emotional context signals than others (i.e., arousal).

It is important to note that while music provides a useful tool for eliciting dynamic emotions, it has some confounds that may constrain our interpretations. For instance, while our music stimuli may possess less intrinsic conceptual overlap with the item memoranda than do emotional images or scenes from films or video clips, people can still generate complex semantic content, such as narratives and mental imagery, while listening to unfamiliar instrumental music[92–94]. Because participants were instructed to form mental narratives using the items during encoding, there might have been unaccounted features of the music or images that promoted successful narrative encoding and thereby influenced temporal memory. To address this possibility, future studies should test how linguistic content generated while listening to music or other dynamically evocative media relates to various aspects of memory.

Critically, we used music to manipulate the temporal stability of emotional contexts, enabling us to test how emotions alter the processing and encoding of an otherwise neutral series of events. In this way, the music/neutral-image pairing is somewhat similar to episodic associative learning paradigms, whereby neutral images are paired with appetitive or aversive unconditioned stimuli[95]. One key advantage of this approach is that it prevents an explicit induction of emotion during memory retrieval, which could trigger encoding specificity effects. Using neutral memoranda also largely avoids confounding the effects of emotion on encoding and retrieval processes, which is challenging to disentangle in memory paradigms that use IAPS images or still frames from films as memoranda. Additionally, everyday emotions often alter the way we process information that is intrinsically neutral (e.g.[96]).

It is important to acknowledge that the strengths of our experimental manipulation may also limit the generalizability of our findings to temporal memory in different emotional situations. Many emotional experiences contain rich visual and semantic information, and emotional stimuli themselves often become the most vivid and memorable content of such events. For instance, emotions are often elicited by perceptual and semantic features intrinsic to the stimulus itself (e.g., the flash of a weapon or the sound of an explosion)[39]. Memories of traumatic events, including intrusive flashbacks, often involve sensory information that is causally relevant and/or conceptually related to salient low-level features of aversive events[97]. Furthermore, everyday experiences are often narrative in nature[98]. They also involve the ongoing construal of complex information, such as causal relationships, character histories, and motives[99]. Additionally, the conceptual overlap between an emotion-eliciting stimulus and neutral memoranda may also dictate whether certain neutral details will be subsequently remembered or forgotten[51]. The intense, elevated arousal induced by aversive images or sounds could uncover the role of emotional arousal in shaping the structure of memory. Considering these findings, future research is needed to determine whether the memory effects we observed also emerge under more naturalistic conditions and whether they contain traces of the emotion-inducing stimulus.

In conclusion, we found that music-induced emotion dynamics were related to event segmentation effects in episodic memory. Changes in emotion also influenced the memorability of individual neutral items and their temporal contexts after a delay. Consideration of dynamic emotion-memory interactions, including models of how memories are adaptively organized to guide future behaviors, may help to inform treatments for psychopathologies characterized by deficient event segmentation and/or memory disorganization, such as PTSD[4]. For example, positive music may offer a safer and more pleasant way to restore the original temporal context of traumatic memories, which are typically highly fragmented and decontextualized[97]. The current results also hold promise for innovative future research investigating if manipulating emotional states can enhance the salience and memorability of neutral information in both educational and clinical settings. Ultimately, it will be important to examine if emotional-related interventions lead to positive mental health outcomes, in part by improving the temporal coherence of emotionally significant memories.

## Methods

### OSF pre-registration
The experimental design, musical stimuli, *Emotion Compass* design, power analyses, and all reported analyses were preregistered through the Open-Science Framework (Preregistration date: November 15, 2021). The preregistration can be accessed via the first author's OSF profile: https://doi.org/10.17605/osf.io/s8g5n. All source code for the *Emotion Compass* and experiment as well as the musical stimuli are also available on the first author's OSF profile: https://doi.org/10.17605/osf.io/s8g5n. There were no deviations from planned exclusion criteria or preregistered analyses. Any additional analyses are specified as non-preregistered and exploratory throughout the manuscript.

### Data collection
**Power analysis.** To estimate the total number of participants needed in this study, we assumed a moderate effect size for a GLM with our predictors of interest (F-squared = 0.2) based on the effect size derived from a general linear model with two predictors in a similar emotional event boundary experiment[22] (boundary arousal and item event position; GLM model comparison: w = 0.59). This power analysis indicated a sample size of 51. However, we assumed that online data collection would yield more noise in our sample than traditional lab-based memory experiments, so we increased our target sample size to 80.

**Participants.** Accounting for up to 20% attrition in online recruitment and/or failure to pass memory performance exclusion criteria, ninety-six healthy young adult participants (mean age = 27 yr old, SD = 4.9 yr; 61 females) were recruited online using Prolific, a website that connects researchers with eligible participants based on a wide range of

inclusion criteria. Eligibility criteria included: (1) age of 18 to 35 years old; (2) United States or Canada nationality; (3) English first language; (4) normal or corrected-to-normal vision; (5) no history of serious head injuries (e.g., concussion); (6) previously participation in 50 studies on Prolific; and (6) a minimum 90% approval rating on Prolific (percentage of studies for which the participant has been approved by the experimenter). The memory experiment and music rating task was hosted by Pavlovia and built-in PsychoPy[100]. Survey data was collected via Qualtrics. Informed consent was obtained in accordance with procedures approved by the University of California, Los Angeles Institutional Review Board. Participants were compensated $9/h for study participation. Neither sex nor ethnicity, race, or other socially relevant groupings were considered in the experimental design, and that data was self-reported by participants on a questionnaire.

Nine participants (n = 9) did not complete the Day 1 task. An additional 6 participants (n = 6) that completed the Day 1 task did not meet inclusion criterion of 50% accuracy or above on temporal memory performance, leaving eighty-one participants for all Day 1 temporal memory analyses (n = 81). Of those eighty-one participants, seventy-two completed all Day 2 memory tasks (n = 72) and were included in all Day 2 memory analyses. All experimental protocols were approved by the University of California at Los Angeles Institutional Review Board.

## Experimental Paradigm

**Image stimuli.** The image stimuli consisted of 360 images of everyday objects displayed on a white background (300 × 300 dpi). The images were selected from the DinoLab Object Database (https://mariamh. shinyapps.io/dinolabobjects/). Selected images excluded categories that could be construed as highly arousing or of extreme valence, such as food, military equipment, and animals.

**Music stimuli.** To develop non-linguistic emotional stimuli, we hired three film score composers, who were all graduate students at New York University's Film Composition program. Each composer wrote original pieces of music using four voices/instruments (violin, guitar, cello, and piano). The composers were asked to divide the pieces into segments, where each segment conveyed a specific emotion. The length of each section and the timing of the transition from one emotion segment to another were left up to the composer, with the requirement that each segment be no less than 30 s. Additionally, the tempo and tonal center (key) were set to be the same across all emotional segments. In this way, emotions were evoked by overlapping acoustic features, allowing us to partially disentangle the effects of musical changes from felt emotional changes on memory.

Emotional categories were selected based on dimensions identified from a large, cross-cultural musical corpus in which people reported both dimensional and categorical emotional responses. After discussion with the composers regarding what was feasible given the musical constraints, we chose music from five distinct emotional categories: sad/depressing, anxious/tense, calm/relaxing, joyous/cheerful, and dreamy/nostalgic. Composers wrote 7 different musical segments for each emotion category, using different musical elements during each recapitulation to promote distinguishability across the segments. For the purposes of this study, we chose emotional events designed to convey both extremes of the arousal and valence dimensions: anxiety (high arousal, low valence), sadness (low arousal, low valence), joy (high arousal, high valence), and calm (low arousal, high valence), and thus did not include any dreamy/nostalgic clips.

These events were spliced into 10 novel "songs", each consisting of 3 emotional segments, separated by a 6–9 s musical transition period. Each song was spliced such that they did not contain more than 2 segments of the same level of valence and arousal. Thus, all 10 songs included notable transitions into segments of differing valence and arousal. See Supplementary Table 1 for timestamps of the emotional events and information about the inter-segment transitions within each song.

**Depression and trauma symptom scales.** Prior to performing the music sequence encoding task, participants completed the Beck Depression Inventory (BDI)[101], the PTSD Checklist Criterion A (PCL)[102], and the Persistent and Intrusive Negative Thoughts Scale (PINTS)[103]. We were specifically interested in individual differences in depression and trauma symptomatology correlating with the usage of the *Emotion Compass*. These results were not central to the aims of this experiment and will be reported elsewhere.

**Item sequence encoding.** For each image sequence, participants viewed a series of 24 random object images while listening to one of the songs in the background. Each image was presented in the center of a gray background for 3 s. The image order was completely randomized for each subject. During the inter-stimulus interval (ISI) between each object, a white fixation cross was displayed in the middle of the screen for 2 s. To promote deep encoding and encourage temporal processing, participants were instructed to form a mental narrative as they viewed the images. As an attention check, participants also had to press the spacebar as quickly as possible whenever they incorporated the image into their mental narrative. Each participant viewed a total of 11 item sequences accompanied by a unique musical composition. Prior to the start of the first trial of each sequence, the song played for 5 s to avoid potential orienting effects to music listening during encoding. The first sequence served as a practice block in which participants experienced a full encoding-retrieval block. This sequence always included the same "dreamy" musical composition. The practice sequence enabled participants to become accustomed to the encoding and memory tasks. Data from the practice block was not included in the final analyses. The order of all non-practice songs was randomized across participants.

**Delay distractor task.** To create a 45-s study-test delay and reduce potential recency effects, participants performed an arrow detection task after studying each item sequence. In this phase, a rapid stream of either left-facing (<) or right-facing (>) arrow symbols appeared in the middle of the screen for 0.5 s each. These arrow screens were separated by 0.5-s ISI screens with a centrally presented black fixation cross. Participants had to indicate which direction the arrow was pointing via button press as quickly as possible.

**Day 1 temporal memory tests.** Following each block of the sequence encoding task, participants were tested on two aspects of episodic memory: temporal order and temporal distance. Participants were shown 10 pairs of objects from the prior sequence, one pair at a time. We queried temporal order memory by having participants indicate which of two probe items from the prior sequence had appeared first (primacy decision). After this choice, the same pair of items remained on-screen, and participants had to rate the temporal distance between the two items. For the temporal distance rating, participants rated item pairs as having appeared 'very close', 'close', 'far', or 'very far' apart from each other in the prior sequence. Crucially, each pair of items was always presented with three intervening items during encoding, thereby always having the same objective distance. Thus, any differences in temporal distance ratings between the item pairs are completely subjective. The order of item pairs as well as the screen position of each item (left or right) was completely randomized (average screen presentation order: 50.2% correct order, 49.8% incorrect order). There was an 8-s time limit for each response and the screen advanced when a button was pressed. Participants who performed numerically worse than the statistical chance level (50%) on the temporal order test were removed from the Day 1 temporal memory analyses (n = 6). After these

exclusions, a sample of n = 81 remained for all Day 1 memory analyses. See Supplementary Software 1 for the code for the Day 1 tasks.

**Emotion compass task.** After the entire music encoding task was completed, participants listened to each song again in the same order as encoding. During this time, participants provided real-time estimates of their moment-to-moment emotional reactions to each musical clip using the *Emotion Compass* (see Fig. 1). The *Emotion Compass* is a 2D animated circumplex grid, inspired by the circumplex model of emotion[104], that continuously tracks participants' subjective valence (positive vs negative: x-axis) and arousal (high vs low: y-axis) reactions to the musical clips using cursor feedback. Participants were instructed to control the position of a "planchette" circle using their computer cursor. The color of the planchette changed according to the cursor position along the x-axis (positive valence = green; negative valence = red) and y-axis (high arousal = high saturation; low arousal = low saturation) within the *Emotion Compass* at each frame. All participants practiced using the *Emotion Compass* to a non-tested musical clip before providing ratings for the tested musical clips. Prior to the initialization of the *Emotion Compass*, the musical composition played for 5 s to avoid potential orienting effects on music listening. Participants were explicitly instructed to rate their "natural felt emotional reaction to the music, rather than what [you] perceive the music to be." To ensure compliance with this instruction, participants completed a survey on their *Emotion Compass* usage at the end of the study. Specifically, participants were asked: "On the scale below, please rate whether you mainly tracked how the music sounded versus how the music made you feel", on a 5-point Likert scale of: "Completely how the music sounded (1)" to "Completely how the music made me feel (5)". All participants used in the analyses reported using the *Emotion Compass* to track "Completely how the music made them feel (5)". See Supplementary Software 1 for the code of the *Emotion Compass*.

**Day 2 item recognition and temporal source memory tests.** Approximately 24 h after the Day 1 session (average = 22 h and 36 min), participants were presented with all object images from the Day 1 encoding session as well as 120 new images (lure rate = 50%) in a randomized order. For each trial, participants indicated whether they saw the image on Day 1 and rated their confidence in their judgment ('definitely old'; 'maybe old', 'definitely new'; 'maybe new'). If the participant endorsed an item as "OLD", their memory was then tested for temporal source or position in its 24-item list. Temporal source ratings were made using a "temporal slider" divided into 24 units, where each unit reflected the serial position of each item in a list (the leftmost point referring to the beginning of the sequence and the rightmost point referring to the end of the sequence). Of the 87 participants who completed the first day of the experiment, 72 returned for the delayed memory tests. All these participants passed the exclusion criteria and were included in the Day 2 memory analyses. See Supplementary Software 1 for the code for Day 2 tasks.

**Emotion compass preprocessing**
**Emotion compass pre-processing.** Filtering the *Emotion Compass* data consisted of two steps: 1) mean-filtering the data by averaging across every 3 frames (sampling from 720 frames per song to 240; 6 HZ to 2 HZ); 2) applying a Savitzky-Golay filter (*window* = 50; *order* = 5) to the mean-filtered subject-wide profiles. We chose to use a Savitzky-Golay filter because this technique can filter over high-frequency bands (i.e., a low-pass filter). It can also simultaneously capture sudden changes in low-frequency bands[105,106], which were more likely to reflect veridical changes in an emotional context. We then manually inspected each participant's valence and arousal profile for each song and filtered out participants who did not pass the inclusion criteria for correct usage of the *Emotion Compass*.

To qualify for appropriate usage of the compass tool, participant-wise profiles for valence and arousal time-series ratings had to pass the following criteria: 1) indicate engagement by changing valence and arousal >50% of the song; 2) indicate a meaningful signal by not jumping across the valence or arousal axis >2 times within a 3 s window; 3) avoid frenetic, nonsensical movement that was likely an idiosyncratic artifact of online cursor tracking. A total of twenty-two participants did not meet these criteria, resulting in a sample of n = 65 for the *Emotion Compass* profile analyses.

To retain otherwise well-performing participants in our analyses, we decided to use song-level rather than subject-level emotion ratings. To create normative temporally dynamic emotional profiles for each song, we concatenated valence and arousal scores across all cleaned and resampled participant ratings. We then mapped the valence and arousal profiles of each song back onto their respective time-locked image sequences from encoding. The temporal window for each trial was defined as the beginning of the image presentation (3 s) until the end of its following ISI (2 s; total time = 5 s). Thus, each trial from encoding was assigned an average valence and arousal score from its temporally aligned position within its specific valence and arousal profile.

**Emotional change-point detection.** To detect discrete changes in subjectively experienced valence and arousal during music listening, we performed change-point analyses on the valence and arousal profiles for each song (see[107] for an application of change-point analyses to cardiorespiratory data; or[108] for an application of change-point analyses to recognition memory data) using the Python package ruptures[109]. A change-point analysis is used to detect a significant change in the mean and the slope of time-ordered data. Based on an evaluation of change-point algorithms showing that binary segmentation outperforms other change-point algorithms for univariate data signals[110], all models used the binary segmentation algorithm provided by ruptures. Because we designed each song to include three distinct emotional segments (e.g., joy-sad-anxious), we had a strong a priori rationale that each song would elicit 2-to-3 discrete emotional changes in participants' ratings with the compass. Thus, we selected the ruptures default penalization factor (penalization factor = 15), which detected an average of 2.1 change-points across all valence profiles and 2.4 change-points across all arousal profiles. We then extracted the timestamps of the change-points for the valence and arousal profiles of each song. Finally, we mapped these timestamps back onto both Day 1 paired temporal memory and Day 2 delayed memory trials. For Day 1 paired temporal memory trials, this time-window spanned the beginning of the first item in a pair until the end of the ISI of the second item in a pair. For Day 2 memory trials, this time-window spanned the beginning of an item until the end of its subsequent ISI. Trials for which a valence or arousal change-point fell within their encoding time-window were considered "valence boundary" or "arousal boundary" trials, respectively.

**Music segmentation**
**Perceptual boundary annotations.** A separate sample of participants (*n* = 6) was recruited in-lab to annotate the musical, or perceptual boundaries, of the songs. Based on prior event boundary annotation literature[57], participants were instructed to listen to each song and press the spacebar when they perceived the composition of the song to be meaningfully changing. Changes were described as points in the music when there was a major change in pitch, tempo, or tone. Furthermore, annotators were told that each change should be spaced by roughly 30 to 70 s, and that there would be no more than 2-3 major changes in each composition. Based off previous work[57], the similarity among observers was measured using Dice's coefficient, which is the number of matching boundaries (boundaries within 2 trials of each other) divided by a mean number of boundaries. Boundary annotations were highly overlapping, with a Dice's coefficient of 0.823 and an average of 2.04 boundaries per song. Because of the high similarity of

annotations, we then created a "consensus" perceptual boundary annotation by averaging the boundary timestamps across all 6 annotators.

## Mixed-effects models of memory metrics

**Linear mixed models.** To test our key hypotheses that emotional music influences the temporal structure of memory and the memorability of item information, we performed linear mixed modeling analyses (for distance ratings and temporal source outcomes) and generalized linear mixed modeling analyses (for temporal order memory and recognition success outcomes) using the lme4 packages in R[111]. Song-level *Emotion Compass* profiles were mapped back onto and aligned with participant-level sequence encoding data. The valence and arousal ratings of these profiles were then modeled as fixed effects in our linear mixed models and generalized linear mixed models. For the discrete event segmentation (change-point) analyses, item pairs that spanned a change-point were coded as a "1", and items that did not span a change-point were coded as a "0". For analyses of absolute and signed emotion context change, the absolute and signed difference between the to-be-tested item pairs was modeled as the fixed effects, respectively.

For Day 2 memory analyses, the average valence and arousal scores for each item were modeled as fixed effects. For recognition analyses, low-confidence ("Definitely Old" or "Maybe Old") responses for old items were considered hits (hits = 1; misses = 0). For temporal displacement analyses, temporal displacement was calculated by taking the absolute difference between the slider placement and the original encoding index for all hits. All models included random intercepts for participants as well as song identity.

For all analyses, the statistical significance of the regression model was determined by performing a likelihood ratio test for the full model against a model which included all independent variables except for the effect of interest. For analyses including categorical fixed effects, linear mixed-effects, and generalized linear mixed-effects models were followed up with two-tailed post-hoc contrasts of marginal means with FDR corrections to the P-values for multiple comparisons.

**Testing statistical assumptions.** For all linear mixed models, statistical assumptions were tested using the DHARMa package in R[112]. We performed Kolmogorov-Smirnov tests of normality and confirmed that the distribution of residuals did not significantly deviate from a normal distribution (all $p$s > 0.15). For models with categorical predictors, we performed Levene tests of homogeneity of variances and confirmed that the homogeneity of variance assumption was supported (all $p$s > 0.55).

## Reporting summary

Further information on research design is available in the Nature Portfolio Reporting Summary linked to this article.

## Data availability

Source data associated with the figures, the raw data generated in this study, and all music and image stimuli have been deposited in the first author's OSF profile (https://doi.org/10.17605/osf.io/s8g5n)[113]. Source data are provided with this paper.

## Code availability

This experiment used code available in Python and R. Code for analyses is available on the first author's OSF profile. All code for the *Emotion Compass* and the experiment have been made available on the first author's OSF profile (https://doi.org/10.17605/osf.io/s8g5n)[113].

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

## Acknowledgements

We would like to thank Ringo Huang, Jamie Greer, and the ACME lab RAs for their help annotating the musical stimuli. We thank Jesse Rissman, Mary Vitello, and Samuel E Cooper for feedback on analyses. We would also like to thank Alexandra Cohen and Alan Castel for feedback on earlier versions of the manuscript. This work was supported by the National Science Foundation Graduate Research Fellowship Program to M. M.

## Author contributions

M.M., M.S., and D.C. conceived and designed the experiments. M.S. designed the music stimuli. M.M. and D.C. designed the Emotion Compass. M.M. performed the experiments. M.M. analyzed the data. M.M. drafted the paper. M.M., M.S., and D.C. edited the paper.

## Competing interests

The authors declare no competing interests.
