## [Peer Review File · Nature Communications]

Dynamic emotional states shape the episodic structure of memoryREVIEWER COMMENTS

Reviewer #1 (Remarks to the Author):

The paper "Dynamic emotional states shape the episodic structure of memory" by Mason McClay, Matthew E. Sachs and David Clewett investigates how emotional dynamics influence memory separation or integration.

To this aim, they asked participants to study neutral items (object images) while being exposed to emotionally characterized music excerpts. Main findings show that fluctuations in music-driven emotion (and specifically emotional valence) regulate memory integration and separation, thus showing novel evidence that memory is organized around emotional states.

The paper is overall very well written. I found the research question timely, the methodological approach original, and the main findings statistically supported. In general, I think this contribution can provide new insights and perspectives to the study of music cognition and memory dynamics.

In my opinion, some main issues need to be addressed, and some points clarified before publication.

- Authors claim that the emotional valence/arousal manipulated in this study can describe fluctuations in participants' internal world. However, music is an external stimulus and musical emotion can be theoretically approached both in terms of recognition and induction. In the case of recognition, the emotional stimulus can be perfectly described but not necessarily (and internally) felt. This can be the case for simple and unfamiliar melodies (as the ones employed in this study). Can the authors confirm that the emotions were actually induced in participants? Which were the specific instructions (with regard to the music) provided to participants during the object encoding task and the Emotional Compass one? For future studies, psychophysiological measures can also be particularly useful to this purpose.

- If I've well understood, the data-driven event boundaries resulting from the Emotion Compass are employed to create normative temporally-dynamic emotional profiles for each song and are obtained by averaging arousal/valence data from participants ratings. However, it is not clear to me what is the interindividual variability of emotion compass dynamics (if any). In other words, could one argue that the temporal dynamics of participant X better describe his/her memory performance than the averaged dynamics? Also, which is the correspondence between discrete and data-driven continuous boundaries?

- At first reading, I've struggled in understanding the difference between musical and arousal/valence boundaries. Also, it is unclear why a small sample of N=6 annotators was chosen to define the musical boundaries, and how they overlap with the emotional ones. I would therefore suggest 1) to provide more evidence about the existence of boundaries and their differences, that could be supported by acoustic analysis and/or larger samples and/or resuming figure; 2) to refer to musical boundaries as "perceptual" rather "musical" (as musical includes by definition also the emotional aspects); and 3) to define boundaries and what they represent earlier in the text (e.g. p.7).

- Authors interpret the valence effect as the "pleasantness of the music" eliciting memory separation change. However, the positive valence of the music can be only partially related to a pleasantness effect, as pleasurable responses can be also elicited by negatively valenced music. In this sense, a continuous measure of subjective pleasure would have helped in disentangling the possible mechanisms underlying the effect of valence on memory segmentation.

- As the study employs musical material, participants' musical expertise should be reported and, if available, considered in the analyses.

- The music-memory link is studied both in terms of musical memory and music-driven memory for non-musical associated information. I'm curious to know whether authors would expect similar results if testing memory for the music itself. I think, for example, that the -now missing- effect of arousal fluctuations on memory separation/integration could show up if testing the memory for the musical excerpts.

Minor

- fig. 1F the 1-10 scale for the delayed surprise test is not described in the caption – please clarify
- fig. 5C I think there is something wrong with the phrasing in the caption
- P.18 "we also found that the memorability of individual neutral items and their temporal landmarks after a delay ..." Something missing?

Reviewer #2 (Remarks to the Author):

This manuscript by McClay, Sachs, and Clewett aimed to understand the role of emotional event boundaries on episodic memory. In particular, the study used an innovative paradigm in which music varying in emotional valence and arousal served as an encoding context for arbitrary streams of images. In sum, the authors show that negative swings in emotional valence promote event segmentation, whereas other evidence suggests that positive emotional swings promote integration. In general, and somewhat surprisingly (to me, at least), shifts in valence produced far more pronounced effects than shifts in arousal, though both positively influenced item and temporal memory.

Overall, I quite liked this experiment and paper. The manipulation is very clever, and both the musical manipulation and the "emotion compass" metric are novel and very interesting. I do not have any particularly huge criticisms of the work, though I do have a few issues and asks for clarification that I think could stand to be addressed. I will list my issues/concerns below, roughly in order of the paper.

1. I would drop mention of therapeutics or applications of this work from the Abstract. This is perfectly fair game for speculation in the Discussion, however. (On that note, it is not clear what "restoring" memory organization means exactly.)
2. In the Introduction, the authors state: "Yet, surprisingly little work has examined how emotion shapes important temporal aspects of memory." This is a bit overstated in my opinion. In addition to some older studies (e.g., Elizabeth Phelps' lab), the authors themselves cite work from Daniella Palumbo's lab. There is also a recent TICS review by Wang et al. (2022) that delves into this. I would dial this down a bit.
3. In the Introduction, the authors state: "Eliciting dynamic emotions through music has several key advantages over more traditional manipulations of emotional memory. Emotional images and words possess greater semantic relatedness with each other compared to non-emotional stimuli thereby posing a challenge for dissociating emotion's distinct influence on memory." I see the point being made here, but I am not sure music as a manipulation really gets you safely out of the territory of semantic influences and conceptual overlap. A person can easily layer semantic meaning onto music, even music without lyrics. Some would reasonably argue that emotions themselves carry conceptual associations and semantic links among one another. I would agree with the authors that things are perhaps slightly disentangled via their manipulation, but I think this point should be qualified by some caveats.
4. Looking at Figure 2, the compass data struck me as being extremely noisy. I was concerned in my read that the "Calm" music did not look decidedly positive, and the "Anxious" and "Sad" music did not

look decidedly negative (Fig. 2C). Moreover, arousal in general seems to be extremely noisy and does not obviously map on in an expected way (Fig. 2D). My skepticism here was mitigated a bit by looking at Supplemental Figure 3, so frankly I would suggest bringing this into the main paper. Nonetheless, even from that figure, it is clear that the valence and arousal ratings are not as I would expect. For example, the "Sad" music has an above-zero average valence rating. Finally, it appears that differences in valence are far larger in magnitude than differences in arousal, which may or may not relate to the relatively weaker effects of arousal shifts compared to valence shifts in terms of event boundary effects. This was probably my biggest sticking point with the paper, and as a reader, I would like to see these apparent quirks in the data discussed.

5. For Figure 3, unless I've misunderstood something, panels A and B seem to be reversed in the text with regards to the presentation in the figure.

6. In the Results, the authors state: 'We found that under negative states, larger decreases in valence led to worse temporal order memory and more detailed estimates of time (Supplementary Figure 6). Unless I have totally overlooked something, this is not evaluated statistically, and Supplementary Figure 6 does not really provide any real clarity. I think a much clearer and stronger case for this particular result should be made.'

7. In the text, the reference to Supplementary Figure 9 confused me - is this supposed to refer to Supplementary Figure 10?

8. I don't understand why there are connecting lines between the means in Figure 5. These are not continuous data but are rather totally separate recognition measures by category.

9. I struggled for some time to understand Figures 6B and 6D, and how these data related to the rest of the data (A,C) in the figure. This plot/result and its meaning/significance could be explained more clearly.

Reviewer #3 (Remarks to the Author):

I am reviewing the paper titled "Dynamic emotional states shape the episodic structure of memory" by McClay, Sachs & Clewett. Here, the authors used a creative method to elicit changes in emotion (including positive and negative states) and evaluate how those shape temporal (distance and order) memory for neutral items encoded in sequences—in addition to examining item and temporal context (source memory) for those items 24h later. The authors find that 'absolute' as well as negative shifts in valence are associated with event segmentation (as measured by greater temporal distance ratings and lower temporal order memory accuracy). Moreover, they find that shifts in valence and arousal, as well as event boundaries produced by perceptual shifts, all enhance item and temporal source memory for neutral objects; moreover, subjective valence and arousal interact to predict temporal source/context memory, such that high arousal positive states enhance and high arousal negative states impair it.

Overall, I found this is a novel and refreshing approach to study a topic of rapidly growing interest—emotion-temporal memory interactions—and I also found the manuscript incredibly well written. Below follow a few questions and suggestions for analysis that I hope might help fine tune and strengthen an already impressive paper.

1. I understand why the authors would want to begin their emotional boundary analysis by examining 'absolute' shifts in valence (change point) as a first pass analysis; however, when drilling down to unveil the drivers of the emotional boundary effect and testing the impact of continuous (changes in) emotion, I am not sure whether it is warranted or interpretable to continue to use an 'absolute' metric

of valence changes—i.e., one which collapses across ‘negative to positive’ and ‘positive to negative’ shifts, which are expected to produce qualitatively distinct effects based on the prior literature; and indeed they do (re: dilation vs. compression). Moreover, ‘arousal’ is often highly correlated with absolute valence metrics (also called extremity)—thus, it is unclear what information the ‘absolute’ metric analysis yields in Figure 4 (beyond what was already shown in Figure 3)—and perhaps more importantly, whether that information is interpretable (given that valence is definitionally a continuum from negative to positive). Indeed, Figure 4 (bottom: signed valence changes) reveals distinct (opposing) effects of shifting to positive vs. negative valence, which raise the question of whether they should be collapsed into an absolute score. (A suggestion here would be to move that into Supplementary).

2. Analyses reported in Supplementary Figure 6 are important as they reveal whether Figure 4 (bottom) findings of temporal compression/dilation are driven specifically by positive valence shifts within positive or within negative experiences—or whether they are driven by both. However, these analyses should be reported more clearly in the main text (Page 10). For instance, in the text, it is only mentioned that shifts toward lower valence within negative experiences—i.e., shifts toward ‘more negative’ moments—were associated with lower temporal order memory and temporal dilation.

However, the Figure legend of Supplementary Figure 6 states:

“A. When both items in a pair are encoded in a negative or positive state, an increase in valence is related to more subjective temporal compression. B. When both pairs are negative, an increase in valence is related to enhanced temporal order memory, but not when both pairs are positive.”

Regarding (A), if it is the case that a temporal dilation effect (with decreasing valence) is present both within negative and positive experiences, that should be mentioned in the main manuscript—at present, only the effect within negative is mentioned.

In addition, presumably an increase in valence within a negative experience follows a previous reduction in valence—given that that is the key emotional provocation (i.e. NEG: valence reduction). Thus, isn’t it more parsimonious to state that reductions in valence (which negative experiences produce) impair temporal memory? (As opposed to saying that “increases in valence” during negative states improve temporal order memory...?)

3. The authors make a good argument for the advantages of using music as the emotion-inducing stimuli, but it is unclear whether principles learned here generalize to temporal memory for emotional events themselves. This should be more explicitly mentioned in the discussion.

4. Do the authors analyze the interaction between event boundaries produced by perceptual shifts in music vs. by emotional state changes? From the text, it appears that these two factors were analyzed without the interaction term; this should be clarified as those may yield multiplicative effects.

5. Does the type of ‘emotional valence boundary’ (Fig 5) matter for item and temporal context memory LTM effects? I.e., were the effects the same regardless of whether the emotional valence boundary was created by a shift from negative (vs. by a shift to negative)?

6. The finding that temporal context memory was poorer for events encoded during high arousal negative experiences (vs low arousal negative) is important and related to predictions in recent models (e.g. Talmi’s work, Palombo & Cocquyt (2020) TICS; Wang, Tambini Lapate (2022) TICS); this finding should be mentioned/more explicitly contextualized in the discussion.

Minor:

It is interesting that arousal changes did not seemingly produce event boundaries. I wonder whether that is because those arousal changes were necessarily embedded within distinct emotional valence states in the current paradigm. Have the authors entered both (continuous) valence and (change in) arousal factors in a simultaneous or hierarchical mixed effects model to evaluate whether 'controlling for valence' (or allowing valence * arousal factors to interact) reveals a similar or different picture than what is reported in the manuscript?

Page 4: 'normative emotions, such as sadness and joy.' – that is a strange term (normative is often used to refer to emotions outside of psychopathology, which would be difficult to definitively state here), do the authors mean basic emotions, or emotional states often encountered in everyday life? (I'd recommended rephrasing it)

Page 7: The authors state that "To control a priori musical contexts, we had professional musical composers write songs that conveyed four different emotional themes (joyous, anxious, sad, and calm) and then combined them into a series of unique compositions." Can the authors elaborate on how this controls for "a priori musical contexts"? I feel like this sentence might be better placed in the Methods unless it can be further clarified succinctly (it broke the flow of the manuscript).

Page 15: Figure 6 was hard to follow with the small font size and relatively cramped figures; perhaps consider splitting that single-row 4 panel image into a 2 x 2 panel for readability.

Nature Communications Response to Reviewers

Reviewer #1 Summary. *The paper “Dynamic emotional states shape the episodic structure of memory” by Mason McClay, Matthew E. Sachs and David Clewett investigates how emotional dynamics influence memory separation or integration.*

To this aim, they asked participants to study neutral items (object images) while being exposed to emotionally characterized music excerpts. Main findings show that fluctuations in music-driven emotion (and specifically emotional valence) regulate memory integration and separation, thus showing novel evidence that memory is organized around emotional states.

The paper is overall very well written. I found the research question timely, the methodological approach original, and the main findings statistically supported. In general, I think this contribution can provide new insights and perspectives to the study of music cognition and memory dynamics. In my opinion, some main issues need to be addressed, and some points clarified before publication.

Comment #1: Authors claim that the emotional valence/arousal manipulated in this study can describe fluctuations in participants’ internal world. However, music is an external stimulus and musical emotion can be theoretically approached both in terms of recognition and induction. In the case of recognition, the emotional stimulus can be perfectly described but not necessarily (and internally) felt. This can be the case for simple and unfamiliar melodies (as the ones employed in this study). Can the authors confirm that the emotions were actually induced in participants? Which were the specific instructions (with regard to the music) provided to participants during the object encoding task and the Emotion Compass one? For future studies, psychophysiological measures can also be particularly useful to this purpose.

We thank Reviewer 1 for pointing out this important distinction between recognition and induction. During the object encoding task, we explicitly instructed participants to “listen to the music and allow your emotional response to the music to unfold naturally. Allow yourself to be immersed in the music as much as possible while you view the images.” During the Emotion Compass task following the encoding/retrieval tasks, we gave participants the following instruction:

“Your goal in this phase of the experiment is to indicate on the Emotion Compass how the music makes you feel in terms of its positivity/negativity and the intensity of your emotional reaction. Importantly, please do your best to rate how the music makes you feel rather than how you perceive the music to be.”

To ensure task compliance, participants filled out a survey regarding various aspects of their experience using the Emotion Compass following completion of the study. Specifically, we asked:

“On the scale below, please rate whether you mainly tracked how the music sounded versus how the music made you feel”, on a 5-point Likert scale of: “Completely how the music sounded (1)” to “Completely how the music made me feel (5)”.

Out of the participants who passed inclusion criteria for Emotion Compass usage, all of them reported using the Emotion Compass to track “Completely how the music made them feel (5)”. We have now included the instructions for music listening during both the object encoding task as well as the Emotion Compass task in the Methods section on Page 22.

“Participants were explicitly instructed to rate their “natural felt emotional reaction to the music, rather than what [you] perceive the music to be”. To ensure compliance with this instruction, participants completed a survey on their Emotion Compass usage at the end of the study. Specifically, participants were provided with the following instruction: “On the scale below, please rate whether you mainly tracked how the music sounded versus how the music made you feel”, on a 5-point Likert scale of: “Completely how the music sounded (1)” to “Completely how the music made me feel (5)”. All participants used in the analyses reported using the Emotion Compass to track “Completely how the music made them feel (5).”

We definitely agree that including psychophysiological measures, such as dynamic facial expressions, skin conductance, and/or pupillometry, would help validate affect-tracking measures acquired by the Emotion Compass. Alternatively (but equally interesting), it could be the case that objective physiological measures do not track with subjective self-reports of felt emotions. One possibility is that such a dissociation could help explain why we did not observe the expected arousal effects that have been reported using pupillometry measures (e.g., Clewett et al., 2022). Including such physiological measures will be an important and compelling direction for future research on how emotional states influence memory.

Comment #2: If I’ve well understood, the data-driven event boundaries resulting from the Emotion Compass are employed to create normative temporally-dynamic emotional profiles for each song and are obtained by averaging arousal/valence data from participants ratings. However, it is not clear to me what is the interindividual variability of emotion compass dynamics (if any). In other words, could one argue that the temporal dynamics of participant X better describe his/her memory performance than the averaged dynamics? Also, which is the correspondence between discrete and data-driven continuous boundaries?

The question of whether there are differences between the ability of group-level vs. participant level emotion profiles to explain variance in memory performance is very important. We apologize for creating any confusion regarding preprocessing the Emotion Compass ratings. Reviewer 1 is correct that the analyses were all conducted with normative emotional profiles averaged across all participants’ ratings.

We decided to use song-level rather than subject-level emotional profiles for two reasons. First, 22 participants did not pass inclusion criteria for their Emotion Compass usage as described on Page 21:

“1) indicate engagement by changing valence and arousal >50% of the song; 2) indicate a meaningful signal by not jumping across the valence or arousal axis >2 times within a 3 second window; 3) avoid frenetic, nonsensical movement that was likely an idiosyncratic artifact of online cursor tracking.”

These exclusions were likely due to the quality of online data collection. Importantly, using song-level profiles (i.e., group-level) allowed us to include high-performing subjects in the emotion-memory analyses, even if their Compass data was noisy and/or did not pass our preprocessing criteria.

Second, when comparing linear mixed models with song-level and subject-level emotional profile predictors, models with song-level emotional profiles perform better than models with subject-level emotional profiles. Below we report the AIC's for the different model fits:

song-level |

Temporal order ~ absolute valence + absolute arousal; AIC = 9164

Temporal order ~ signed valence + signed arousal; AIC = 9165

Temporal distance ~ absolute valence + absolute arousal; AIC = 19098

Temporal distance ~ signed valence + signed arousal; AIC = 19102

Subject-level |

Temporal order ~ absolute valence + absolute arousal; AIC = 9169

Temporal order ~ signed valence + signed arousal; AIC = 9168

Temporal distance ~ absolute valence + absolute arousal; AIC = 19107

Temporal distance ~ signed valence + signed arousal; AIC = 19108

Having said that, we agree that acquiring meaningful subject-level Emotion Compass ratings is important and may even be associated with individual differences in other relevant measures, such as depression and anxiety symptomatology. We have added the current methodological limitation to the Discussion on Page 18 as a possible interpretation for a lack of arousal effects on segmentation and integration in Day 1 pairwise memory testing:

“However, because we collapsed emotion ratings across participants to account for variability in online cursor tracking and to retain otherwise well-performing participants on the memory tests, there is the possibility that the song-level arousal ratings were not sensitive enough for our Day 1 memory analyses.”

Regarding the second comment about the correspondence between discrete and data-driven continuous boundaries, we explored corresponding change in valence and arousal at different

boundary types. Specifically, in **Supplementary Figure 5** (Page 6 of Supplementary Materials) we show how large the shifts in valence and arousal ratings were for all trial pairs spanning each discrete boundary type and their corresponding non-boundary trial pairs. These results show that the amount of emotional change was greatest for the corresponding boundary types. Namely, valence changes were greatest between valence boundary trial pair items, arousal change was greatest for arousal boundary trial pair items, and there was comparatively little valence or arousal change between perceptual boundary pair items. Thus, **Figure S5** illustrates discrete change-points indeed correspond with the magnitude of emotional change.

Comment #3: At first reading, I've struggled in understanding the difference between musical and arousal/valence boundaries. Also, it is unclear why a small sample of N=6 annotators was chosen to define the musical boundaries, and how they overlap with the emotional ones. I would therefore suggest 1) to provide more evidence about the existence of boundaries and their differences, that could be supported by acoustic analysis and/or larger samples and/or resuming figure; 2) to refer to musical boundaries as “perceptual” rather “musical” (as musical includes by definition also the emotional aspects); and 3) to define boundaries and what they represent earlier in the text (e.g. p.7).

We thank Reviewer 1 for recommending these clarifications about “musical” boundaries. Our sample of six annotators was chosen based on prior studies that have used separate groups of raters to annotate event boundaries (e.g., Baldassano, et al., 2017; Chen et al., 2017; Silva, Baldassano, & Fuentemilla, 2019). These studies typically employ ~4-6 human annotators to identify meaningful event boundaries.

The perceptual boundaries for our music stimuli were also validated in a separate publication by the second author, who used a larger sample of raters (N = 40; Sachs, Ochsner, Kozak & Baldassano, 2023). It is noteworthy that, like the current manuscript, that study demonstrated that perceptual changes in music are distinguishable from emotional changes.

To measure the consistency of perceptual boundary ratings across annotators, we report a statistic for inter-rater reliability, namely Dice's coefficient. Our Dice coefficient was relatively high compared to other studies using naturalistic stimuli (Dice's coefficient = .823; Page 23). We also underscore that our music stimuli were composed to have clear, regular musical events ranging from 30-60s long, which allowed us to induce multiple perceptual boundaries within each item sequence.

For clarity, we underscore the difference between perceptual and emotional boundaries, and refer to another study conducted by the second author that shows clear differences between perceptual versus emotional change (Sachs, Ochsner, Kozak & Baldassano, 2023; Page 7). We also now refer to “musical boundaries” as “perceptual boundaries”, or moments when individuals perceive a “meaningful change” in the music itself, throughout the manuscript. This can include inferred emotional changes in the music, but should not refer to any changes to a participant's internal states, which would refer to change in valence and arousal.

“To identify musical, or auditory perceptual, event boundaries driven by acoustic changes in the music, we used a standard event segmentation approach where a separate group of annotators (n = 6) listened to each song and pressed a button whenever they perceived a meaningful perceptual change in the music. A consensus perceptual boundary annotation was then constructed from these data by taking the averaged boundary timepoints across annotators.

The timestamps of these event boundaries were then aligned to the timeline of object presentations during the sequence encoding task. We then sorted the to-be-tested object pairs according to whether they spanned (“boundary-spanning” pair) or did not span (“non-boundary” pair) a perceptual (i.e., musically-induced), valence, or arousal boundary (**Figure 1E**).”

Comment #4: Authors interpret the valence effect as the “pleasantness of the music” eliciting memory separation change. However, the positive valence of the music can be only partially related to a pleasantness effect, as pleasurable responses can be also elicited by negatively valenced music. In this sense, a continuous measure of subjective pleasure would have helped in disentangling the possible mechanisms underlying the effect of valence on memory segmentation.

We thank Reviewer 1 for raising this important point that pleasantness and positive valence are not the same thing. We agree that “pleasantness” is not the most appropriate word to use when describing music that induces positive valence, as unpleasant music can indeed be pleasurable. For this reason, we specifically asked participants to rate how positive versus how negative they felt when listening to the music to avoid making assumptions about discrete emotion categories (e.g., Sachs et al., 2015). We have revised the language in the manuscript about emotional ratings from the Emotion Compass to be more precise (**Page 8**):

“These findings suggest that emotional change, specifically in the experienced positive affect evoked by the music, elicits memory separation effects beyond mere perceptual change.”

Based on a previous study that compared different ways of asking emotional responses to music, we believe that measuring subjective pleasure (from unpleasant to pleasant) would have yielded very similar results to the Emotion Compass. Furthermore, using a continuum labeled by the adjectives “positive” to “negative” has been used interchangeably with “pleasant” to “unpleasant” in previous work (Eerola & Vuoskoski, 2011).

Additionally, in a previous study conducted by the second author, researchers collected continuous ratings of “enjoyment” (as a proxy for measuring pleurability) of music. These enjoyment ratings yielded highly variable results across participants, leading the authors to conclude that a valence scale ranging from the adjectives “negative” to “positive” might be more interpretable for most participants (Sachs et al., 2020).

Comment #5: As the study employs musical material, participants' musical expertise should be reported and, if available, considered in the analyses.

We did not collect ratings of expertise in this sample. However, we would like to note that by using entirely novel musical compositions, we tried to reduce the potential influence of musical expertise on emotional states, event segmentation, and the organization of memory.

Comment #6: The music-memory link is studied both in terms of musical memory and music-driven memory for non-musical associated information. I'm curious to know whether authors would expect similar results if testing memory for the music itself. I think, for example, that the - now missing- effect of arousal fluctuations on memory separation/integration could show up if testing the memory for the musical excerpts.

Reviewer 1's raises an extremely interesting point that testing musical memory could help reveal the effects of arousal on memory organization. We believe one of the strengths of our study is that music is used to induce emotional states that color the processing and encoding of otherwise neutral stimuli. Thus, we were better able to isolate the effects of felt emotions on memory independently of the nature of the memoranda themselves (e.g., all neutral, everyday items). Orienting individuals to process and encode the music itself (if they knew that's what they would be tested on) would likely affect participants' ability to encode the objects, perhaps distracting them away from the task at-hand. Another challenge to researching musical stimuli as the memoranda is that there would be a risk of reinstating arousal/valence during retrieval. This idea emphasizes another important strength of the current study: by using neutral item pairs, we are better able to isolate the effects of felt emotions on encoding processes.

Having said all this, Reviewer 1 raises a very intriguing question that is worthy of future research. There could also be much therapeutic potential of music-related memory in terms of using emotional music to improve memory function in individuals with different disorders of emotion, such as depression, or who experience significant memory decline (e.g., aging or AD).

One challenge to studying the organization of musical memory is creating an appropriate memory test to probe memory integration and separation effects. One could imagine using clips/excerpts of the musical pieces, much like the item pair manipulation used in this study. But the perceptual features of the musical clips themselves could potentially hint at the correct temporal order of those pieces of music. One could, however, examine how emotional pieces of music influence temporal distortions in subjective temporal distance ratings.

There are some indications of how emotions might influence temporal memory from prior work on the effects of arousal on prospective time estimation. For instance, musically-evoked arousal has been shown to relate to temporal dilation using a temporal bisection task (Droit-Volet et al., 2013). While the relationship between prospective and retrospective temporal estimation is unclear (Tsao et al., 2022), there is consistent evidence that participants judge more arousing stimuli as being longer than standard durations during temporal bisection tasks (Lake et al., 2016).

In addition to temporal bisection tasks, an event enumeration task (i.e., counting the number of musical notes; e.g., Ongchoco & Scholl, 2020) could be adapted to test how valence and arousal relate to event segmentation processes. Based upon previous findings in the emotion and time perception literature (Droit-Volet et al., 2013), we expect that arousal would result in a subjective expansion, or dilation of time, in memory. By contrast, a shift away from negative valence music would likely result in temporal compression. In terms of enumeration, increases in valence (i.e., becoming less negative) may promote immersion in a phrase or segment of music, which may result in stronger binding of discrete subordinate events (e.g., notes) together. This immersion effect could reduce the number of counted/enumerated events by orienting individuals towards processing individual items or bits of information at the expense of processing the broader musical sequence.

Comment #7: fig. 1F the 1-10 scale for the delayed surprise test is not described in the caption – please clarify

We apologize for any confusion. The delayed temporal memory test actually used a scale ranging from 1-24, which refers to the 24 objects and their positions within a given song/list of items. We have clarified this in the **Figure 5A** caption (Page 13) with the following language:

“Temporal source memory accuracy was measured by temporal displacement, which is defined as the absolute difference between the endorsed item position on the slider (out of 24 possible positions) and the original item position.”

Comment #8: fig. 5C I think there is something wrong with the phrasing in the caption - P.18 “we also found that the memorability of individual neutral items and their temporal landmarks after a delay ...” Something missing?

We thank Reviewer 1 for pointing this out. We have now fixed this typo by adding: “...was related to discrete perceptual and emotional change as well as positive, arousing states during encoding.”

Responses to Reviewer #2

Reviewer #2 Summary. *This manuscript by McClay, Sachs, and Clewett aimed to understand the role of emotional event boundaries on episodic memory. In particular, the study used an innovative paradigm in which music varying in emotional valence and arousal served as an encoding context for arbitrary streams of images. In sum, the authors show that negative swings in emotional valence promote event segmentation, whereas other evidence suggests that positive emotional swings promote integration. In general, and somewhat surprisingly (to me, at*

least), shifts in valence produced far more pronounced effects than shifts in arousal, though both positively influenced item and temporal memory.

Overall, I quite liked this experiment and paper. The manipulation is very clever, and both the musical manipulation and the "emotion compass" metric are novel and very interesting. I do not have any particularly huge criticisms of the work, though I do have a few issues and asks for clarification that I think could stand to be addressed. I will list my issues/concerns below, roughly in order of the paper.

Comment #1: I would drop mention of therapeutics or applications of this work from the Abstract. This is perfectly fair game for speculation in the Discussion, however. (On that note, it is not clear what "restoring" memory organization means exactly.)

We agree with Reviewer 2's suggestion to remove discussions about therapeutic applications from the Abstract. We have slightly revised our discussion to explore the potential therapeutic implications of the findings, given that these ideas are more speculative. The new final sentence in the Abstract is:

"The rise and fall of emotions thereby have the power to sculpt unfolding experiences into memories of meaningful events."

Comment #2: In the Introduction, the authors state: "Yet, surprisingly little work has examined how emotion shapes important temporal aspects of memory." This is a bit overstated in my opinion. In addition to some older studies (e.g., Elizabeth Phelps' lab), the authors themselves cite work from Daniella Palumbo's lab. There is also a recent TICS review by Wang et al. (2022) that delves into this. I would dial this down a bit.

We completely agree with Reviewer 2 that there has been important work on emotion and temporal memory. We have tempered our language and now state (Page 3):

"By comparison, less work has examined how emotion shapes important temporal aspects of memory."

Comment #3: In the Introduction, the authors state: "Eliciting dynamic emotions through music has several key advantages over more traditional manipulations of emotional memory. Emotional images and words possess greater semantic relatedness with each other compared to non-emotional stimuli thereby posing a challenge for dissociating emotion's distinct influence on memory." I see the point being made here, but I am not sure music as a manipulation really gets you safely out of the territory of semantic influences and conceptual overlap. A person can easily layer semantic meaning onto music, even music without lyrics. Some would reasonably argue that emotions themselves carry conceptual associations and semantic links among one another. I would agree with the authors that things are perhaps slightly disentangled via their manipulation, but I think this point should be qualified by some caveats.

Reviewer 2 raises excellent points about how music can be imbued with meaning. There is indeed evidence showing that unfamiliar music can generate spontaneous narrative imagination (e.g., McAuley et al., 2021; Margulis et al, 2022, PNAS; Margulis et al, 2022, Cognition). We chose musical stimuli because they likely engage semantic processing to a lesser degree than more commonly used “naturalistic stimuli”, such as linguistic narratives or movies. But we agree that semantics and prior knowledge can’t be fully decoupled from musical stimuli, even non-lyrical, unfamiliar music. Furthermore, work on thematic arousal shows that emotionality throughout a narrative can enhance memory for different semantic details of those narratives (Laney et al., 2004). We have revised our description about semantics in the Introduction on Page 4:

“Music, on the other hand, can reliably induce a range of emotions spanning the valence-arousal space with less explicit linguistic structure. Unfamiliar pieces of instrumental music may help prevent individuals from relying on prior knowledge to scaffold episodic encoding processes. Finally, different emotional reactions to music can be evoked by shared acoustic features, such as tone or tempo, enabling some level of control over the lower-level effects of perceptual change on memory encoding. Altogether, these features suggest that music is an effective method for studying the link between emotion dynamics and memory, because it can reduce the influence of semantic relatedness, familiarity, and perceptual change on memory.”

Additionally, we describe some caveats to using music in the Discussion on Page 19:

“It is important to note that while music provides a useful tool for eliciting dynamic emotions, it has some confounds that may constrain our interpretations. For instance, while our music stimuli may possess less intrinsic conceptual overlap with the item memoranda than do emotional images or scenes from films or video clips, people can still generate complex semantic content, such as narratives and mental imagery, while listening to unfamiliar instrumental music^{95–97}. Because participants were instructed to form mental narratives using the items during encoding, it is possible that unaccounted features of the music or images may have facilitated successful narrative encoding and thereby influenced temporal memory. To address this possibility, future studies should test how linguistic content generated while listening to music or other dynamically evocative media relates to various aspects of memory.”

Comment #4: Looking at Figure 2, the compass data struck me as being extremely noisy. I was concerned in my read that the "Calm" music did not look decidedly positive, and the "Anxious" and "Sad" music did not look decidedly negative (Fig. 2C). Moreover, arousal in general seems to be extremely noisy and does not obviously map on in an expected way (Fig. 2D). My skepticism here was mitigated a bit by looking at Supplemental Figure 3, so frankly I would

suggest bringing this into the main paper. Nonetheless, even from that figure, it is clear that the valence and arousal ratings are not as I would expect. For example, the "Sad" music has an above-zero average valence rating. Finally, it appears that differences in valence are far larger in magnitude than differences in arousal, which may or may not relate to the relatively weaker effects of arousal shifts compared to valence shifts in terms of event boundary effects. This was probably my biggest sticking point with the paper, and as a reader, I would like to see these apparent quirks in the data discussed.

We thank Reviewer 2 for their helpful comments regarding how well valence and arousal map onto the respective basic emotion categories. First, we appreciate that the kernel density plots are relatively difficult to interpret (**Figure 2C** and **Figure 2D**). We clarify that the data in **Figure 2C** and **Figure 2D** is primarily used to verify that individuals are using the Emotion Compass appropriately and that their subjective ratings map onto these basic emotions. However, this was simply meant to validate our novel affect-tracking tool. A key strength of our manipulation is that we are measuring continuous valence and arousal changes, which: 1) provides a high degree of sensitivity for tracking the ebb and flow of felt emotions, and 2) obviates the need to make assumptions about what categorical emotion someone is experiencing. In essence, the goal of these Figures is simply to show that people are using the Emotion Compass arbitrarily and that their ratings are meaningful.

Furthermore, we want to apologize and clarify that Supplementary Figure 5 shows the differences in valence and arousal between boundary and non-boundary item pairs. We accidentally reference **Supplementary Figure 3** instead of the correct **Supplementary Figure 5** at the end of **Page 7**. We now have revised this reference to be to **Supplementary Figure 5**.

Indeed, when looking at valence and arousal change across item pairs broken down by boundary vs. non-boundary trial types irrespective of collapsed categories, there is greater arousal change at arousal change-points than there is valence change at valence change-points (see **Supplementary Figure 5**). As displayed in this figure, there is a large amount of arousal change at arousal change-points. We believe this provides convincing evidence that the lack of a relationship between arousal and memory cannot be explained by limited changes in arousal.

Further, we found that arousal ratings were meaningfully related to delayed memory outcomes: participants showed better item recognition and temporal source memory for arousal change-points than non-change-points (see **Figure 5**). We also found that item-locked valence and arousal interact in predicting recognition and temporal source memory, such that high valence and high arousal states were related to better item and source memory than low valence, low arousal states. Taken together, these findings reveal that arousal had a much stronger influence on memory after a delay (especially for items and temporal source measures), which is consistent with prior work on emotional memory (e.g., Sharot and Phelps, 2004; Yonelinas and Ritchey, 2015).

Regarding Reviewer 2's comment concerning sad states having above-zero valence, it is common in music-emotion literature for music that conveys a negative emotion state

(particularly sadness) to induce positive emotions in listeners. This has been shown for both sad and angry music (Sachs et al., 2015; Droit-Volet et al, 2013). Thus, it is not surprising that listeners in this study rated the sad pieces as inducing a positive valence and provides further evidence that participants were indeed rating how they felt rather than their perception of the emotionality of the music itself. Importantly, as mentioned above, our analyses were agnostic to the *a priori* emotion categories. Our goal was to leverage measures of continuous linear change in the emotion profiles for the songs and then relate these to memory measures for all trial pairs.

We definitely agree that clarifying these issues and explaining our logic to the reader would be very helpful. We have fixed the typo to refer to **Supplementary Figure 5** and discuss differences in valence and arousal change at respective timepoints on **Page 7**:

“We then used a data-driven approach to identify emotional valence and arousal boundaries in felt emotions during music listening. Specifically, we conducted a change-point analyses on the valence and arousal timeseries for each song acquired from the *Emotion Compass*. The change-point algorithm identified when there was a significant change in the mean and slope of the valence and arousal ratings timeseries, separately (**Figure 2B**). The timestamps of these event boundaries were then aligned to the timeline of object presentations during the sequence encoding task. Object pairs that were to be tested for temporal order and distance memory were then labeled according to whether they spanned (“boundary-spanning” pair) or did not span (“non-boundary” pair) a perceptual, valence, or arousal boundary (**Figure 1E**). As illustrated in the arousal and valence ratings in **Supplementary Figure 5**, the change-point algorithm appropriately labeled the different valence and arousal boundary types.”

We also now further elaborate on the patterns of variability of valence and arousal variability in the Discussion on Page 18:

“We observed comparable variability in both valence and arousal ratings across music listening. Further, we found that item pairs that span valence and arousal change-points coincide with prominent changes in valence and arousal, respectively, verifying that the Compass was labeling different emotional shifts appropriately. Participants also provided emotion ratings that aligned with the predefined emotion categories of the songs designed by the composers. For these reasons, it is unlikely that the null arousal effect was driven by participants being unable to simultaneously track changes in both valence and arousal during music listening. However, because we collapsed emotional ratings across participants to account for variability in online cursor tracking and to retain otherwise well-performing participants, there is the possibility that the song-level arousal ratings were not sensitive enough to account for variance in temporal order and temporal distance memory.”

Comment #5: For Figure 3, unless I've misunderstood something, panels A and B seem to be reversed in the text with regards to the presentation in the figure.

We thank Reviewer 2 for detecting this error in the main text. We have revised the text to correctly reference panels A and B.

Comment #6: In the Results, the authors state: 'We found that under negative states, larger decreases in valence led to worse temporal order memory and more detailed estimates of time (Supplementary Figure 6). Unless I have totally overlooked something, this is not evaluated statistically, and Supplementary Figure 6 does not really provide any real clarity. I think a much clearer and stronger case for this particular result should be made.'

We agree that evaluating this statistically would strengthen our case that movement within a certain emotional space matters for segmentation/integration. We have now updated **Figure 4** to include these critical results. This new panel in Figure 4 provides a breakdown of the memory-valence correlations by which valence "state" (or hemisphere of the Compass) participants were in. We also include the accompanying statistical results on **Page 10**:

"Because this was a linear effect, it is somewhat unclear if integration was driven by emotions moving from more to less negative states (staying within the left hemisphere of the Emotion Compass and drifting rightward toward the y-axis; see **Figure 4D**) or moving from less positive to more positive states (staying within the right hemisphere but drifting rightwards and away from the y-axis). To determine the nature of these signed valence effects, we conducted separate regressions for item pairs encoded during purely positive versus purely negative states (i.e., both items were associated with negative or positive emotion). We found that under negative states, larger increases in valence led to more compressed estimates of time ($\beta = -.047$, standard error (SE) = .01, $\chi^2(1) = 7.36$, $p = .006$; **Figure 4E, top panel**) as well as enhanced temporal order memory ($\beta = .12$, standard error (SE) = .04, $\chi^2(1) = 5.36$, $p = .008$; **Figure 4F, top panel**). By contrast, moving from a less positive to more positive state was not related to temporal distance ratings ($\beta = -.013$, standard error (SE) = .009, $\chi^2(1) = 1.41$, $p = .16$; **Figure 4E, bottom panel**) or temporal order memory ($\beta = -.01$, standard error (SE) = .003, $\chi^2(1) = 1.3$, $p = .93$; **Figure 4F, bottom panel**). These findings suggest that memory integration is driven by a shift away from highly negative emotional states but not a shift from relatively low positive states to more positive states."

Importantly, we have revised our general theoretical approach and interpretation of this analysis to be framed around memory integration rather than segmentation. That is, we based our interpretation on what it means to move from left-to-right on the compass as opposed to the other direction. This also aligns these analyses with the original signed valence results in which we

found that increasing valence was related to better memory integration (i.e., temporal compression and enhanced temporal order memory).

We also believe this framing has important therapeutic implications, as it suggests that *reductions in negative affect* can benefit the temporal integration of memories. On the other hand, adding more positive affect on top of an already positive state has no significant benefit for memory integration.

Comment #7: In the text, the reference to Supplementary Figure 9 confused me - is this supposed to refer to Supplementary Figure 10?

We thank Reviewer 2 for catching this error. We indeed intended to refer to **Supplementary Figure 10** here, not Supplementary Figure 9. But because we have deleted supplementary Figure 6, the appropriate reference here is the new **Supplementary Figure 9** (previous Supplementary Figure 10). We have now updated this in the main text on **Page 13**.

Comment #8: I don't understand why there are connecting lines between the means in Figure 5. These are not continuous data but are rather totally separate recognition measures by category.

We have now removed these connecting lines.

Comment #9: I struggled for some time to understand Figures 6B and 6D, and how these data related to the rest of the data (A,C) in the figure. This plot/result and its meaning/significance could be explained more clearly.

We thank Reviewer 2 for the opportunity to revise and clarify **Figure 6**. We have now split the panels into separate rows to help reduce the amount of clutter. We also realized that the “more accurate” and “less accurate” labels for panels C and D were reversed. When temporal displacement is lower, that is indicative of more accurate temporal source memory. We have now fixed this in the figure. Furthermore, we agree that it would be helpful to clarify what we mean by temporal source memory and temporal displacement. We have now included the following clarifications on **Page 14**:

“Here, temporal source memory accuracy was measured by temporal displacement, which is defined as the absolute difference between the endorsed item position on the slider (out of 24 possible positions) and the original item position in a list. Thus, lower temporal displacement values are indicative of less error, or better temporal source memory.”

Responses to Reviewer #3

Reviewer #3 Summary. *I am reviewing the paper titled “Dynamic emotional states shape the episodic structure of memory” by McClay, Sachs & Clewett. Here, the authors used a creative method to elicit changes in emotion (including positive and negative states) and evaluate how those shape temporal (distance and order) memory for neutral items encoded in sequences—in addition to examining item and temporal context (source memory) for those items 24h later. The authors find that ‘absolute’ as well as negative shifts in valence are associated with event segmentation (as measured by greater temporal distance ratings and lower temporal order memory accuracy). Moreover, they find that shifts in valence and arousal, as well as event boundaries produced by perceptual shifts, all enhance item and temporal source memory for neutral objects; moreover, subjective valence and arousal interact to predict temporal source/context memory, such that high arousal positive states enhance and high arousal negative states impair it.*

Overall, I found this is a novel and refreshing approach to study a topic of rapidly growing interest—emotion-temporal memory interactions—and I also found the manuscript incredibly well written. Below follow a few questions and suggestions for analysis that I hope might help fine tune and strengthen an already impressive paper.

Comment #1: I understand why the authors would want to begin their emotional boundary analysis by examining ‘absolute’ shifts in valence (change point) as a first pass analysis; however, when drilling down to unveil the drivers of the emotional boundary effect and testing the impact of continuous (changes in) emotion, I am not sure whether it is warranted or interpretable to continue to use an ‘absolute’ metric of valence changes—i.e., one which collapses across ‘negative to positive’ and ‘positive to negative’ shifts, which are expected to produce qualitatively distinct effects based on the prior literature; and indeed they do (re: dilation vs. compression). Moreover, ‘arousal’ is often highly correlated with absolute valence metrics (also called extremity)—thus, it is unclear what information the ‘absolute’ metric analysis yields in Figure 4 (beyond what was already shown in Figure 3)—and perhaps more importantly, whether that information is interpretable (given that valence is definitionally a continuum from negative to positive). Indeed, Figure 4 (bottom: signed valence changes) reveals distinct (opposing) effects of shifting to positive vs. negative valence, which raise the question of whether they should be collapsed into an absolute score. (A suggestion here would be to move that into Supplementary).

We thank Reviewer 3 for this suggestion and think that this is an extremely important point. We believe that it is important to keep the absolute valence and arousal change analyses in the main text for three reasons. First, while being somewhat redundant to the change-point boundary analysis reported in **Figure 3**, the absolute continuous change analysis is a simpler measure of emotional change than the changepoint analysis, and thus is important to demonstrate as a meaningful predictor of segmentation. Second, showing that absolute change is also a predictor of memory helps to validate the Emotion Compass as a subjective emotional tracking tool. Thirdly, while we think this is a very valid suggestion, we also want to adhere to the absolute change analyses that were proposed in our pre-registration.

We nevertheless agree that the more interesting and appropriate effects of interest are the *type* of valence change. Therefore, we now highlight these findings more explicitly in the main text. Specifically, we added panels D, E, and F to **Figure 4** that show the differences in memory separation/integration effects (otherwise reported in **Supplementary Figure 6**) when participants shift from a more-to-less negative state compared to moving from a less-to-more positive state. The following revised text has now been included on **Page 10**:

“Because this was a linear effect, it is somewhat unclear if integration was driven by emotions moving from more to less negative states (staying within the left hemisphere of the Emotion Compass and drifting rightward toward the y-axis; see **Figure 4D**) or moving from less positive to more positive states (staying within the right hemisphere but drifting rightwards and away from the y-axis). To determine the nature of these signed valence effects, we conducted separate regressions for item pairs encoded during purely positive versus purely negative states (i.e., both items were associated with negative or positive emotion). We found that under negative states, larger increases in valence led to more compressed estimates of time ($\beta = -.047$, standard error (SE) = .01, $\chi^2(1) = 7.36$, $p = .006$; **Figure 4E, top panel**) as well as enhanced temporal order memory ($\beta = .12$, standard error (SE) = .04, $\chi^2(1) = 5.36$, $p = .008$; **Figure 4F, top panel**). By contrast, moving from a less positive to more positive state was not related to temporal distance ratings ($\beta = -.013$, standard error (SE) = .009, $\chi^2(1) = 1.41$, $p = .16$; **Figure 4E, bottom panel**) or temporal order memory ($\beta = -.01$, standard error (SE) = .003, $\chi^2(1) = 1.3$, $p = .93$; **Figure 4F, bottom panel**). These findings suggest that memory integration is driven by a shift away from highly negative emotional states but not a shift from relatively low positive states to more positive states.”

Comment #2A: Analyses reported in Supplementary Figure 6 are important as they reveal whether Figure 4 (bottom) findings of temporal compression/dilation are driven specifically by positive valence shifts within positive or within negative experiences—or whether they are driven by both. However, these analyses should be reported more clearly in the main text (Page 10). For instance, in the text, it is only mentioned that shifts toward lower valence within negative experiences—i.e., shifts toward ‘more negative’ moments—were associated with lower temporal order memory and temporal dilation. However, the Figure legend of Supplementary Figure 6 states:

“A. When both items in a pair are encoded in a negative or positive state, an increase in valence is related to more subjective temporal compression. B. When both pairs are negative, an increase in valence is related to enhanced temporal order memory, but not when both pairs are positive.”

Regarding (A), if it is the case that a temporal dilation effect (with decreasing valence) is present both within negative and positive experiences, that should be mentioned in the main manuscript—at present, only the effect within negative is mentioned.

We thank Reviewer 3 for raising the important point of the valence-related effects on memory segmentation. We now report the linear mixed modeling results for this finding, along with new panels D, E, and F to illustrate the decomposition of more-to-less-negative versus less-to-more-positive valence change on temporal memory on Page 10 (see response to Reviewer #3 Comment #1 above).

Importantly, we have revised our general theoretical approach and interpretation of this analysis to be framed around memory integration rather than segmentation. That is, we based our interpretation on what it means to move from left-to-right on the compass as opposed to the other direction. This also aligns these analyses with the original signed valence results in which we found that increasing valence was related to better memory integration (i.e., temporal compression and enhanced temporal order memory).

We also believe this framing has important therapeutic implications, as it suggests that *reductions in negative affect* can benefit the temporal integration of memories. On the other hand, adding more positive affect on top of an already positive state has no significant benefit for memory integration.

Because we now report these findings in Figure 4 of the main text, we have also decided to delete Supplementary Figure 6.

Comment #2B: In addition, presumably an increase in valence within a negative experience follows a previous reduction in valence—given that that is the key emotional provocation (i.e., NEG: valence reduction). Thus, isn't it more parsimonious to state that reductions in valence (which negative experiences produce) impair temporal memory? (As opposed to saying that “increases in valence” during negative states improve temporal order memory...?)

We thank Reviewer 3 for raising this important and helpful point. Importantly, the musical stimuli may start negative (e.g., with a sad or anxious context), and move toward a positive context. Therefore, while it may take time to move toward the initial negative state, the primary change would take place during the transition to the next, more positive, segment. Further, this interpretation is more consistent with the initial signed-change of valence analysis (**Figure 4B and 4C**; see response to Reviewer #3 Comments #1 and #2A above), where we find a main effect of increasing valence related to temporal compression and enhanced temporal order memory. We have now updated our interpretation of this analysis more thoroughly on Page 10 (see response to Comment #1, above).

Comment #3: The authors make a good argument for the advantages of using music as the emotion-inducing stimuli, but it is unclear whether principles learned here generalize to temporal memory for emotional events themselves. This should be more explicitly mentioned in the discussion.

We thank Reviewer 3 for this thoughtful comment and agree that this is an important point to address in the discussion. Prior work demonstrate that neural representations of emotions may

be domain independent (showing overlap for music, non-musical vocal, and visual/movie-induced emotions with fMRI; Kim et al., 2017; Sachs et al., 2018), and there is some evidence music-evoked emotions have similar effects on memory as emotions induced with image stimuli (Talmani et al., 2022). That being said, the degree to which our findings would generalize is a relatively underexplored question.

However, we also interpret our use of musical stimuli to be similar to that of other neutral memoranda-emotional context/outcome pairing. Thereby, we believe we can make similar inferences about generalizability. We clarify this point now in the Discussion (Page 19):

“Critically, we used music to manipulate the temporal stability of emotional contexts, enabling us to test how emotions color and shape memories for an otherwise neutral series of events. In this way, the music/neutral-image pairing is somewhat similar to episodic associative learning paradigms, whereby neutral images are paired with appetitive or aversive unconditioned stimuli⁹³. One key advantage of this approach is that it prevents an explicit induction of emotion during memory retrieval, which could trigger encoding specificity effects. Using neutral memoranda also largely avoids confounding the effects of emotion on encoding and retrieval processes, which is challenging to disentangle in memory paradigms that use IAPS images or still frames from films as memoranda. Additionally, everyday emotions often color the way we process information that is intrinsically neutral (e.g., ⁹⁴). Using background music stimuli to alter one’s emotional states thereby provides a useful analogue for real-world experiences.”

Comment #4: Do the authors analyze the interaction between event boundaries produced by perceptual shifts in music vs. by emotional state changes? From the text, it appears that these two factors were analyzed without the interaction term; this should be clarified as those may yield multiplicative effects.

We thank Reviewer 3 for the suggestion and agree that this is an important possibility. We ran exploratory hierarchical mixed models with an interaction term for discrete perceptual vs. emotional boundaries. There is no significant interaction between perceptual boundaries and valence or arousal boundaries. Moreover, our reported hierarchical models show that valence relates to segmentation even when controlling for perceptual and arousal boundaries. We interpret these findings as revealing that the influence of perceptual and valence boundaries on the organization and segmentation of memory are relatively orthogonal. For Reviewer 3’s interest, we include model output with interaction terms below:

Temporal order ~

Perceptual * arousal boundaries, $\beta = -.02$, standard error (SE) = .024, $\chi^2(1) = -.833$, $p = .4$;

Perceptual * valence boundaries, $\beta = .04$, standard error (SE) = .027, $\chi^2(1) = 1.7$, $p = .08$;

Subjective temporal distance ~

Perceptual * arousal boundaries, $\beta = .009$, standard error (SE) = .008, $\chi^2(1) = 1.188$, $p = .23$;
Perceptual * valence boundaries, $\beta = -.009$, standard error (SE) = .008, $\chi^2(1) = -1.18$, $p = .24$;

Comment #5: Does the type of ‘emotional valence boundary’ (Fig 5) matter for item and temporal context memory LTM effects? I.e., were the effects the same regardless of whether the emotional valence boundary was created by a shift from negative (vs. by a shift to negative)?

This is a very compelling question. We have conducted exploratory analyses investigating how the type of transition – broken down by both high/low valence and high/low arousal, separately – relates to temporal order/distance and delayed recognition and temporal source memory. Our exploratory findings were interesting: we found that temporal context memory was marginally significantly better for items that coincided with boundaries that either went from less-to-more positive states, more-to-less negative states, or signaled a transition from negative to positive states. Notably, this is consistent with our Day 1 temporal memory results, which showed greater temporal compression and enhanced temporal order memory for item pairs that were studied when individuals shifted from more-to-less negative states (i.e., staying in purely negative states, but transitioning to less negative states; see **Figure 4**). For recognition memory, memory was nearly identical for items coinciding with boundaries going from negative to positive and positive to negative states.

We note that the current study was not designed to address the specific question of type of transition ahead of time, because the emotion category transitions were not perfectly balanced across the music stimuli (i.e., joy → sad etc.). These analyses are also underpowered when we split the boundary trials into high/low valence. Although we cannot investigate this idea in the current data, we agree that examining if the type of discrete emotional category shift influences memory segmentation is an intriguing topic that warrants further research.

Comment #6: The finding that temporal context memory was poorer for events encoded during high arousal negative experiences (vs low arousal negative) is important and related to predictions in recent models (e.g., Talmi’s work, Palombo & Cocquyt (2020) TICS; Wang, Tambini Lapate (2022) TICS); this finding should be mentioned/more explicitly contextualized in the discussion.

We agree that these review papers help anchor our study in this growing field of work on emotion and temporal memory, especially with respect to negative arousing states. We now discuss our results in the context of these studies in the Discussion on **Pages 16-17**:

“Interestingly, these effects of valence shifts on memory appeared to be specific to the domain of negative affect. We found that experiencing a shift away from highly negative to less negatively valenced emotional states, rather than from less positive to more positively valence states, enhanced event integration processes in memory³². Conversely, this relationship can also be interpreted as a

drastic shift towards negative states resulting in stronger event segmentation effects in memory. Indeed, this finding accords with evidence that negative stimuli tend to narrow the focus of attention on individual pieces of information, which in turn impairs relational strategies that support encoding temporal associations⁷¹. It also aligns with research showing that more item-focused encoding under neutral (i.e., less negative) conditions disrupts temporal binding processes for temporally extended sequences⁷² as well as work showing that negative emotions disrupt binding and retrieval of temporal context information^{17,20,21}. Importantly, this valence-specific effect suggests that alleviating negative states rather than simply adding more positive valence to an already positive state is critical for preserving memories of the order and structure of distinct events. This finding may have important therapeutic implications for improving the coherence of memory in disorders characterized by intense negative affect and memory disorganization. Individuals who suffer from highly negative emotions may stand to gain the most from interventions aimed at improving one's mood and memory for the order, timing, and causal relationships between meaningful events."

Comment #7: It is interesting that arousal changes did not seemingly produce event boundaries. I wonder whether that is because those arousal changes were necessarily embedded within distinct emotional valence states in the current paradigm. Have the authors entered both (continuous) valence and (change in) arousal factors in a simultaneous or hierarchical mixed effects model to evaluate whether 'controlling for valence' (or allowing valence * arousal factors to interact) reveals a similar or different picture than what is reported in the manuscript?

We thank Reviewer 3 and agree that this is an important possibility. We have run hierarchical mixed effects models to determine if controlling for valence leads to different results. The output from the model shows when controlling for one another, valence and arousal continue to have nearly the same effects on temporal order and subjective temporal dilation. Further, there is no significant interaction between valence and arousal. We interpret these results as evidence that valence is independently driving much of the segmentation processes on Day 1. For transparency, we report these models that include continuous change in valence and arousal and their interaction terms below:

Temporal order ~

Valence change * arousal change, $\beta = .023$, standard error (SE) = .017, $\chi^2(1) = 1.4$, $p = .16$;

Subjective temporal distance ~

Valence change * arousal change, $\beta = .002$, standard error (SE) = .0032, $\chi^2(1) = -1.3$, $p = .14$;

Comment #8: Page 4: 'normative emotions, such as sadness and joy.' – that is a strange term (normative is often used to refer to emotions outside of psychopathology, which would be

difficult to definitively state here), do the authors mean basic emotions, or emotional states often encountered in everyday life? (I'd recommended rephrasing it)

We agree that a more appropriate term would be basic/everyday emotions. We have changed this throughout the text.

Comment #9: Page 7: The authors state that “To control a priori musical contexts, we had professional musical composers write songs that conveyed four different emotional themes (joyous, anxious, sad, and calm) and then combined them into a series of unique compositions.” Can the authors elaborate on how this controls for “a priori musical contexts”? I feel like this sentence might be better placed in the Methods unless it can be further clarified succinctly (it broke the flow of the manuscript).

We thank Reviewer 3 for providing this opportunity to clarify what we meant by “a priori musical contexts” and improve the flow of the manuscript. By a priori contexts, we meant that since we designed the music, we had a template for the musical context changes. We have clarified this section in the main text (Page 7):

“To create music with controlled perceptual and emotional context changes, we had professional musical composers write songs that conveyed four different basic emotional themes (joyous, anxious, sad, and calm). We then mixed the emotional themes into a series of unique 120s compositions. Each composition contained three unique 30-40s emotional segments that were connected by 6-9s musical transition periods. These transitions helped reduce the likelihood that participants would perceive an obvious and abrupt perceptual change between segments (see Supplementary Table 1 for the event structure design of each musical composition). Importantly, the compositions were designed to elicit different patterns of dynamic change across experienced valence and arousal to promote dynamic variability across emotional states within each composition. All compositions included no more than 2 emotional themes of the same valence (positive, negative) or arousal (high, low) across all 3 segments.”

We believe it is helpful to keep these details in the Results, because our main analyses rely on this experimental control over the stimuli based on their composition. Because many readers will go through the manuscript chronologically, we wish to foreground this point early on. We hope this revised language helps clarify this point while preserving the flow of the text. For more details about how we designed and parameterized the music stimuli, please see the Method section (Pages 20-21 under *Music Stimuli*).

Comment #10: Page 15: Figure 6 was hard to follow with the small font size and relatively cramped figures; perhaps consider splitting that single-row 4 panel image into a 2 x 2 panel for readability.

We thank Reviewer 3 for this useful suggestion for editing Figure 6. We have now split the panels into separate rows to help decrease clutter.

References

Clewett, D., Gasser, C., & Davachi, L. (2020). Pupil-linked arousal signals track the temporal organization of events in memory. *Nature Communications*, 11(1), Article 1.

Baldassano, C. *et al.* (2017). Discovering Event Structure in Continuous Narrative Perception and Memory. *Neuron* 95, 709-721.e5.

Chen J, Leong YC, Honey CJ, Yong CH, Norman KA, Hasson U (2017). Shared memories reveal shared structure in neural activity across individuals. *Nat Neurosci* 20:115–125.

Silva, M., Baldassano, C., & Fuentemilla, L. (2019). Rapid Memory Reactivation at Movie Event Boundaries Promotes Episodic Encoding. *The Journal of Neuroscience*, 39(43), 8538–8548.

Sachs, M. E., Ochsner, K., Kozak M. & Baldassano, C. (2023). Brain state dynamics reflect emotion transitions induced by music. *bioRxiv*, 2023-03.

Sachs, M. E., Damasio, A., & Habibi, A. (2015). The pleasures of sad music: a systematic review. *Frontiers in human neuroscience*, 9, 404.

Eerola, T., & Vuoskoski, J. K. (2011). A comparison of the discrete and dimensional models of emotion in music. *Psychology of Music*, 39(1), 18–49.

Sachs, M. E., Habibi, A., Damasio, A., & Kaplan, J. T. (2020). Dynamic intersubject neural synchronization reflects affective responses to sad music. *NeuroImage*, 218, 116512.

Droit-Volet, S., Ramos, danilo, Bueno, L., & Bigand, E. (2013). Music, emotion, and time perception: The influence of subjective emotional valence and arousal? *Frontiers in Psychology*, 4.

Tsao, A., Yousefzadeh, S. A., Meck, W. H., Moser, M.-B., & Moser, E. I. (2022). The neural bases for timing of durations. *Nature Reviews Neuroscience*, 23(11), Article 11.

Lake, J. I., LaBar, K. S., & Meck, W. H. (2016). Emotional modulation of interval timing and time perception. *Neuroscience & Biobehavioral Reviews*, 64, 403–420.

Ongchoco, J. D. K., & Scholl, B. J. (2020). Enumeration in time is irresistibly event-based. *Psychonomic Bulletin & Review*, 27(2), 307–314.

McAuley, J. D., Wong, P. C., Mamidipaka, A., Phillips, N., & Margulis, E. H. (2021). Do you hear what I hear? Perceived narrative constitutes a semantic dimension for music. *Cognition*, 212, 104712.

Margulis, E. H., Wong, P. C., Turnbull, C., Kubit, B. M., & McAuley, J. D. (2022). Narratives imagined in response to instrumental music reveal culture-bounded intersubjectivity. *Proceedings of the National Academy of Sciences*, 119(4), e2110406119.

- Margulis, E. H., Williams, J., Simchy-Gross, R., & McAuley, J. D. (2022). When did that happen? The dynamic unfolding of perceived musical narrative. *Cognition*, 226, 105180.
- Laney, C., Campbell, H. V., Heuer, F., & Reisberg, D. (2004). Memory for thematically arousing events. *Memory & Cognition*, 32(7), 1149–1159.
- Kim, J., Shinkareva, S. V., & Wedell, D. H. (2017). Representations of modality-general valence for videos and music derived from fMRI data. *NeuroImage*, 148, 42-54.
- Sachs, M. E., Habibi, A., Damasio, A., & Kaplan, J. T. (2018). Decoding the neural signatures of emotions expressed through sound. *Neuroimage*, 174, 1-10.
- Kim, J., Shinkareva, S. V., & Wedell, D. H. (2017). Representations of modality-general valence for videos and music derived from fMRI data. *NeuroImage*, 148, 42-54.
- Sachs, M. E., Habibi, A., Damasio, A., & Kaplan, J. T. (2018). Decoding the neural signatures of emotions expressed through sound. *Neuroimage*, 174, 1-10.
- Talamini, F., Eller, G., Vigl, J., & Zentner, M. (2022). Musical emotions affect memory for emotional pictures. *Scientific Reports*, 12(1), 10636.
- Fullana, M. A., Dunsmoor, J. E., Schruers, K. R. J., Savage, H. S., Bach, D. R., & Harrison, B. J. (2020). Human fear conditioning: From neuroscience to the clinic. *Behaviour Research and Therapy*, 124, 103528.
- Maren, S., Phan, K. L., & Liberzon, I. (2013). The contextual brain: Implications for fear conditioning, extinction and psychopathology. *Nature Reviews Neuroscience*, 14(6), Article 6.
- Talmi, D., Lohnas, L. J., & Daw, N. D. (2019). A retrieved context model of the emotional modulation of memory. *Psychological Review*, 126(4), 455-485.
- Palombo, D. J., & Cocquyt, C. (2020). Emotion in Context: Remembering When. *Trends in Cognitive Sciences*, 24(9), 687–690.
- Wang, J., Tambini, A., & Lapate, R. C. (2022). The tie that binds: Temporal coding and adaptive emotion. *Trends in Cognitive Sciences*, 26(12), 1103–1118.

REVIEWER COMMENTS

Reviewer #1 (Remarks to the Author):

The authors precisely answered to all the issues raised in the first review and clarified all the open points, modifying the main text accordingly.
I confirm the positive evaluation of this manuscript and I congratulate the authors for their work.

Reviewer #2 (Remarks to the Author):

The authors have done a nice job of addressing my comments (and, in my opinion, those of the other reviewers as well). I have no further concerns and am happy to endorse the paper for publication.

Reviewer #3 (Remarks to the Author):

I am reviewing the first revision of the paper titled "Dynamic emotional states shape the episodic structure of memory" by McClay, Sachs & Clewett. Overall, the authors did a thorough job of addressing concerns raised during the revision. I have the following remaining concerns:

Results, Page8: All of the 95% CI appear to be missing.

(Related to) R3 Comment #2B: This Reviewer still feels that the current terminology may give rise to confusion. The term "increases in valence" – which turns out (in the current experiment) are only operative as far as sculpting temporal memory during negative (not positive!) states – is rarely (if ever) used to describe reductions of negative affect. Thus, I reckon that the current terminology might obfuscate an otherwise very clear (and compelling!) set of findings. If the authors stick to the current framing, I'd encourage pairing it with a much clearer description of the findings they have in their discussion, e.g.: "We found that experiencing *a shift away from highly negative to less negatively valenced emotional states*... enhanced event integration processes in memory"

Comment #3: While the authors somewhat acknowledged in the Response letter the limitation that the current study doesn't necessary speak to temporal memory for emotional events in an emotional situation—rather, it speaks to temporal memory for neutral events that happened to co-occur with an emotional state—that limitation was not appropriately acknowledged in the Discussion/manuscript (instead, the revised paragraph appears to do the opposite-- as it emphasizes only strengths, without acknowledging any limitations). As per usual, "there is no free lunch"—the current approach is very elegant and creative in that it circumvents some of the methodological concerns associated with rich visual or narrative emotional stimuli, however, many if not most emotional events in everyday life are in fact visual and/or narrative in nature, and traumatic memories often include the original, emotionally-evocative events themselves. Therefore, whether insights garnered from the current study will generalize to temporal memory for emotional events—with their full sensory richness, which is critical in psychopathology—remains to be determined as far as this Reviewer knows.

Response to Reviewers

Reviewer #3

Summary. I am reviewing the first revision of the paper titled “Dynamic emotional states shape the episodic structure of memory” by McClay, Sachs & Clewett. Overall, the authors did a thorough job of addressing concerns raised during the revision. I have the following remaining concerns:

Comment #1: Results, Page 8: All of the 95% CI appear to be missing.

We thank Reviewer 3 for pointing this out! We have now filled in the missing CI's.

(Related to) R3 Comment #2B: This Reviewer still feels that the current terminology may give rise to confusion. The term “increases in valence” – which turns out (in the current experiment) are only operative as far as sculpting temporal memory during negative (not positive!) states – is rarely (if ever) used to describe reductions of negative affect. Thus, I reckon that the current terminology might obfuscate an otherwise very clear (and compelling!) set of findings. If the authors stick to the current framing, I'd encourage pairing it with a much clearer description of the findings they have in their discussion, e.g.: “We found that experiencing *a shift away from highly negative to less negatively valenced emotional states*... enhanced event integration processes in memory”

We thank Reviewer 3 for this very helpful suggestion and agree that clarity and consistency of this terminology is important. We have now updated our description of these memory integration effects in the Results (throughout Page 10) to be more consistent with the interpretation/description in the Discussion. Namely, we describe the correlation effect as examining “a shift from a more negative emotional state to a less negatively-valenced state” as opposed to an “increase in valence”. We also summarize this effect at the end of Page 10:

“These findings suggest that event integration in memory is primarily driven by a shift away from a highly negative emotional state to a less negative emotional state (i.e., a reduction in negative valence specifically).”

Comment #3: While the authors somewhat acknowledged in the Response letter the limitation that the current study doesn't necessary speak to temporal memory for emotional events in an emotional situation—rather, it speaks to temporal memory for neutral events that happened to co-occur with an emotional state—that limitation was not appropriately acknowledged in the Discussion/manuscript (instead, the revised paragraph appears to do the opposite-- as it emphasizes only strengths, without acknowledging any limitations). As per usual, “there is no free lunch”—the current approach is very elegant and creative in that it circumvents some of the methodological concerns associated with rich visual or narrative emotional stimuli, however, many if not most emotional events in everyday life are in fact visual and/or narrative in nature, and traumatic memories often include the original, emotionally-evocative events themselves.

Therefore, whether insights garnered from the current study will generalize to temporal memory for emotional events—with their full sensory richness, which is critical in psychopathology—remains to be determined as far as this Reviewer knows.

We thank Reviewer 3 for raising this important and thoughtful concern again. We strongly agree that there are important differences between emotions elicited by a surrounding context versus emotions elicited by (and/or are intrinsic to) an emotional stimulus or narrative structure. We are hopeful that acknowledging this important limitation has led to a more balanced description of our findings and their contributions to the literature. We have now updated our Discussion (Page 20, Paragraph 1) to further address this important distinction and temper our interpretations:

“It is important to acknowledge that these strengths of our experimental manipulation may also limit the generalizability of our findings to temporal memory in different emotional situations. Many emotional experiences contain rich visual and semantic information, and emotional stimuli themselves often become the most vivid and memorable content of such events. For instance, emotions are often elicited by perceptual and semantic features intrinsic to the stimulus itself (e.g., the flash of a weapon or the sound of an explosion; Talmi & Moscovitch, 2004). Indeed, memories of traumatic events, including intrusive flashbacks, often involve sensory information that is causally relevant and/or conceptually related to salient low-level features of aversive events (Ehlers et al., 2002). Furthermore, everyday experiences are often narrative in nature (Zwaan & Radvansky, 1998). They also involve the ongoing construal of complex information, such as causal relationships, character histories, and motives (Shin & Dubrow, 2021). Additionally, conceptual overlap between an emotion-eliciting stimulus and neutral memoranda may also dictate whether certain neutral details will be subsequently remembered or forgotten (Laney et al., 2004). The intense, elevated arousal induced by aversive images or sounds may also uncover a role of emotional arousal in shaping the structure of memory. Considering these findings, future research is needed to determine whether the memory effects we observed also emerge under more naturalistic conditions and whether they contain traces of the emotion-inducing stimulus.”

References

Talmi, D., & Moscovitch, M. (2004). Can semantic relatedness explain the enhancement of memory for emotional words? *Memory & Cognition*, 32(5), 742–751.

Ehlers, A., Hackmann, A., Steil, R., Clohessy, S., Wenninger, K., & Winter, H. (2002). The nature of intrusive memories after trauma: The warning signal hypothesis. *Behaviour Research and Therapy*, 40(9), 995–1002.

Zwaan, R. A., & Radvansky, G. A. (1998). Situation Models in Language Comprehension and Memory. *Psychological Bulletin*, 123(2), 162–185.

Shin, Y. S., & DuBrow, S. (2021). Structuring memory through inference-based event segmentation. *Topics in Cognitive Science*, 13(1), 106–127.

Laney, C., Campbell, H. V., Heuer, F., & Reisberg, D. (2004). Memory for thematically arousing events. *Memory & Cognition*, 32(7), 1149–1159.

REVIEWERS' COMMENTS

Reviewer #3 (Remarks to the Author):

The authors have done an excellent job thoroughly addressing all of the Reviewers' concerns.

Response to Reviewers

Reviewer #3 (Remarks to the Author):

The authors have done an excellent job thoroughly addressing all of the Reviewers' concerns.

Response: We thank Reviewer 3 for their helpful suggestions and for helping us strengthen our paper.